# Stochastic Gradient Descent-Ascent and Consensus Optimization for Smooth Games: Convergence Analysis under Expected Co-coercivity

**Nicolas Loizou**[*]
Mila and DIRO,
Université de Montréal

**Hugo Berard**
Mila and DIRO,
Université de Montréal

**Gauthier Gidel**[†]
Mila and DIRO,
Université de Montréal

**Ioannis Mitliagkas**[†]
Mila and DIRO,
Université de Montréal

**Simon Lacoste-Julien**[†]
Mila and DIRO,
Université de Montréal

## Abstract

Two of the most prominent algorithms for solving unconstrained smooth games are the classical stochastic gradient descent-ascent (SGDA) and the recently introduced stochastic consensus optimization (SCO) [Mescheder et al., 2017]. SGDA is known to converge to a stationary point for specific classes of games, but current convergence analyses require a bounded variance assumption. SCO is used successfully for solving large-scale adversarial problems, but its convergence guarantees are limited to its deterministic variant. In this work, we introduce the *expected co-coercivity* condition, explain its benefits, and provide the first last-iterate convergence guarantees of SGDA and SCO under this condition for solving a class of stochastic variational inequality problems that are potentially non-monotone. We prove linear convergence of both methods to a neighborhood of the solution when they use constant step-size, and we propose insightful stepsize-switching rules to guarantee convergence to the exact solution. In addition, our convergence guarantees hold under the arbitrary sampling paradigm, and as such, we give insights into the complexity of minibatching.

## 1 Introduction

Motivated from the recent interest in solving adversarial formulations in machine learning such as generative adversarial networks (GANs) [Goodfellow et al., 2014], we consider in this paper a more abstract formulation of the problem and focus on solving the following *unconstrained* stochastic variational inequality (VI) problem:[2]

$$\text{Find} \quad x^* \in \mathbb{R}^d \quad \text{such that} \quad \xi(x^*) = \frac{1}{n} \sum_{i=1}^{n} \xi_i(x^*) = 0, \tag{1}$$

where each $\xi_i : \mathbb{R}^d \to \mathbb{R}^d$ is Lipschitz continuous. Further, we assume that the problem (1) has a unique[3] solution $x^*$ and that the operator $\xi$ is $\mu$-*quasi-strongly monotone*: there is a $\mu \geq 0$ such that:

$$\langle \xi(x), x - x^* \rangle \geq \mu \|x - x^*\|^2 \quad \forall x \in \mathbb{R}^d \tag{2}$$

---

[*]Corresponding author: nicolasloizou1@gmail.com. [†]Canada CIFAR AI Chair.

[2]While our presentation focuses on this finite-sum structure, most of our convergence results can easily be adapted to the general stochastic setting (see App. D). Also, we do not use the full power of variational inequalities that usually have constraints [Harker and Pang, 1990], but standard algorithms for (1) are coming from this literature [Gidel et al., 2018].

[3]This assumption can be relaxed; but for simplicity of exposition we enforce it.

35th Conference on Neural Information Processing Systems (NeurIPS 2021).

If $\mu = 0$, then we say that $\xi$ satisfies the variational stability condition: $\langle \xi(x), x - x^* \rangle \geq 0$ [Hsieh et al., 2020]. In the variational inequality literature, condition (2) is also known as strong stability condition [Mertikopoulos and Zhou, 2019] or as strong Minty variational inequality (MVI) [Diakonikolas et al., 2021, Song et al., 2020].

Problem (1) generalizes the solution of several types of *stochastic smooth games* [Facchinei and Kanzow, 2007, Scutari et al., 2010, Mertikopoulos and Zhou, 2019]. The simplest example is the unconstrained min-max optimization problem (also called a *zero-sum* game):

$$\min_{x_1 \in \mathbb{R}^{d_1}} \max_{x_2 \in \mathbb{R}^{d_2}} \frac{1}{n} \sum_{i=1}^{n} g_i(x_1, x_2), \tag{3}$$

where each component function $g_i : \mathbb{R}^{d_1} \times \mathbb{R}^{d_2} \to \mathbb{R}$ is assumed to be smooth. Here, $\xi_i$ represents the appropriate concatenation of the block-gradients of $g_i$: $\xi_i(x) := (\nabla_{x_1} g_i(x_1, x_2); -\nabla_{x_2} g_i(x_1, x_2))$, where $x := (x_1; x_2)$. Solving (1) then amounts to finding a stationary point $x^* = (x_1^*; x_2^*)$ for (3), which under a convex-concavity assumption for $g_i$ for example, implies that it is a global solution for the min-max problem. More generally, we might seek the pure Nash equilibrium of a $k$-players game, where each player $j$ is simultaneously trying to find the action $x_j^*$ which minimizes with respect to $x_j \in \mathbb{R}^{d_j}$ their own cost function $\frac{1}{n} \sum_{i=1}^{n} f_i^{(j)}(x_j, x_{-j})$, while the other players are playing $x_{-j}$, which represents $x = (x_1, \ldots, x_k)$ with the component $j$ removed. Here, $\xi_i(x)$ is the concatenation over all possible $j$'s of $\nabla_{x_j} f_i^{(j)}(x_j, x_{-j})$.

Such finite sum formulations appear in several machine learning applications such as generative adversarial networks (GANs) [Goodfellow et al., 2014], robust learning [Wen et al., 2014] or even some formulations of reinforcement learning [Pfau and Vinyals, 2016]. A standard algorithm that has been used to solve (1) is the stochastic version of the classical gradient method [Dem'yanov and Pevnyi, 1972, Nemirovski et al., 2009] or its variance reduced version [Balamurugan and Bach, 2016], that we call stochastic gradient descent-ascent (SGDA) in this paper.[4] More recently, Mescheder et al. [2017] analyzed some limitations of the gradient method in the context of GAN training and proposed an alternative efficient algorithm which could be used to solve (1) that they called consensus optimization (CO), which combines gradient updates with the minimization of $\|\xi(x)\|^2$. While the practical version of their algorithm for large $n$ is stochastic (SCO, that randomly samples $i$'s) and displayed good performance [Mescheder et al., 2017], the only global convergence rate guarantees existing in the literature so far is only for the deterministic variant [Azizian et al., 2020, Abernethy et al., 2021].

The classical results from the stochastic VI literature are inappropriate for several reasons. First, a *uniform bound* over $x$ on the variance $\mathbb{E}\left[\|\xi_i(x) - \xi(x)\|^2\right]$ is typically assumed to get convergence guarantees (see e.g. Nemirovski et al. [2009], Gidel et al. [2018], Mertikopoulos and Zhou [2019], Yang et al. [2020], Lin et al. [2020b]), but this is not compatible with the *unconstrained* aspect of (1). For example, suppose $g_i$ is a quadratic function, then the variance typically goes to infinity as $x \to \infty$. More appropriate relaxed assumptions have been considered to prove the convergence of other algorithms for (1) such as the stochastic extragradient method [Hsieh et al., 2020, Mishchenko et al., 2020] and its variance-reduced version [Chavdarova et al., 2019], or the stochastic Hamiltonian gradient method [Loizou et al., 2020], but not yet to the best of our knowledge for SGDA nor SCO. Second, the classical analysis for SGDA [Nemirovski et al., 2009] typically considers the convergence of the *average of the iterates* rather than for the *last-iterate*. However, as pointed out among others by Daskalakis et al. [2018] and Chavdarova et al. [2019], getting last-iterate convergence is important to apply the methods on potentially non-monotone problems such as GANs, where averaging is not appropriate. The only non-asymptotic last iterate convergence result for SGDA that we are aware of is Lin et al. [2020b], which focuses on a different class of problems (not assuming quasi-strong monotonicity) but it relies on strong assumptions on $\mathbb{E}\left[\|\xi_i(x)\|^2\right]$ (see Section 4).

In this paper, we address both of these issues. We generalize the recent improved analysis of SGD [Gower et al., 2019] to the case of unconstrained stochastic variational inequality (1), and prove the last-iterate convergence for both SGDA and SCO without requiring any bounded variance assumption. We focus on quasi-strongly monotone VI problems, a class of structured non-monotone operators for which we are able to provide tight convergence guarantees and avoid the standard issues (cycling and divergence of the methods) appearing in the more general non-monotone regime.

---

[4]We use this suggestive name motivated from the min-max formulation (3), though we also call SGDA the simple update $x^{k+1} = x^k - \alpha \xi_{i_k}(x^k)$ to solve (1) in the more general non-zero sum game scenario.

**Main Contributions.** The key contributions of this work are summarized as follows:

- We propose the *expected co-coercivity* (EC) assumption, which is the appropriate generalization of the expected smoothness assumption from Gower et al. [2019] to Problem (1). We explain the benefits of EC and show that is strictly weaker than the bounded variance assumption and "growth conditions" previously used for the analysis of stochastic algorithms for (1).

- Using the EC assumption, we prove the first last-iterate convergence guarantees for stochastic gradient descent-ascent (SGDA) on (1) without any unrealistic noise assumption. We show a linear convergence rate to a neighborhood of $x^*$ when constant step-size is used, and a $O(1/k)$ rate to the exact solution when using a decreasing step-size rule. For the latter, we propose a theoretically motivated switching rule from a constant to a decreasing step-size to get faster convergence.

- Using the EC assumption, we provide the first convergence analysis of a stochastic variant (SCO) of the consensus optimization (CO) algorithm proposed by Mescheder et al. [2017] and previously used to trained GANs. In particular, we prove last-iterate convergence for SCO, for both constant and decreasing step-sizes. As a corollary of our results, we obtain an improved convergence analysis for the deterministic CO. Furthermore, we explain how the update rule of the stochastic Hamiltonian gradient descent [Loizou et al., 2020] is a special case of the SCO and show that in this scenario, our analysis matches the theoretical guarantees presented in Loizou et al. [2020].

- Inspired by recent results from the optimization literature [Gower et al., 2019], we give the first stochastic reformulation of the variational inequality problem (1) which enables us to provide convergence guarantees of SGDA and SCO under the *arbitrary sampling paradigm* [Richtárik and Takáč, 2016]. This allows us to give insights into the complexity of minibatching.

## 2 Arbitrary Sampling: Stochastic Reformulation of Problem (1)

In this work, we provide theorems through which we can analyze all minibatch variants of the two algorithms under study, SGDA and SCO. To do this, we construct a so-called "stochastic reformulation" of the variational inequality problem (1). Our approach is inspired by recently proposed stochastic reformulations of standard optimization problems, like the empirical risk minimization in Gower et al. [2019] and linear systems in Richtárik and Takác [2020], Loizou and Richtárik [2020a,b].

In each step of our algorithms, we assume we are given access to unbiased estimates $g(x) \in \mathbb{R}^d$ of the operator such that $\mathbb{E}[g(x)] = \xi(x)$. For example, we can use a minibatch to form an estimate of the operator such as $g(x) = \frac{1}{b} \sum_{i \in S} \xi_i(x)$, where $S \subset \{1, \ldots, n\}$ will be chosen uniformly at random and $|S| = b$. To allow for any form of minibatching, we use the *arbitrary sampling* notation $g(x) = \xi_v(x) := \frac{1}{n} \sum_{i=1}^{n} v_i \xi_i(x)$, where $v \in \mathbb{R}_+^n$ is a random *sampling vector* drawn from some distribution $\mathcal{D}$ such that $\mathbb{E}_{\mathcal{D}}[v_i] = 1$, for $i = 1, \ldots, n$. Note that the unbiasedness follows immediately from this definition of the sampling vector: $\mathbb{E}_{\mathcal{D}}[\xi_v(x)] = \frac{1}{n} \sum_{i=1}^{n} \mathbb{E}_{\mathcal{D}}[v_i] \xi_i(x) = \xi(x)$.

Thus, with each user-defined distribution $\mathcal{D}$, we are able to introduce a stochastic reformulation of problem (1) as follows:

$$\text{Find} \quad x^* \in \mathbb{R}^d \quad \text{such that} \quad \mathbb{E}_{\mathcal{D}} \left[ \xi_v(x^*) := \frac{1}{n} \sum_{i=1}^{n} v_i \xi_i(x^*) \right] = 0. \tag{4}$$

Since $\xi_v(x)$ as an unbiased estimate of the operator $\xi(x)$, we can now use stochastic (simultaneous) gradient descent-ascent (SGDA) to solve (4) as follows:

$$x^{k+1} = x^k - \alpha_k \xi_{v^k}(x^k), \tag{5}$$

where $v^k \sim \mathcal{D}$ is sampled i.i.d at each iteration and $\alpha_k > 0$ is a stepsize. We highlight that in our analysis, we allow to select *any* distribution $\mathcal{D}$ that satisfies $\mathbb{E}_{\mathcal{D}}[v_i] = 1 \; \forall i$, and for different selection of $\mathcal{D}$, (5) yields different interpretation as an SGDA method for solving the original problem (1).

In this work, we mostly focus on the $b$–minibatch sampling, however note that our analysis holds for every form of minibatching and for several choices of sampling vectors $v$.

**Definition 2.1** (Minibatch sampling). Let $b \in [n]$. We say that $v \in \mathbb{R}^n$ is a $b$–minibatch sampling if for every subset $S \in [n]$ with $|S| = b$, we have that $\mathbb{P}\left[ v = \frac{n}{b} \sum_{i \in S} e_i \right] = 1/\binom{n}{b} := \frac{b!(n-b)!}{n!}$

By using a double counting argument, one can show that if $v$ is a $b$–minibatch sampling, it is also a valid sampling vector ($\mathbb{E}_{\mathcal{D}}[v_i] = 1$) [Gower et al., 2019]. See Gower et al. [2019] for other choices of sampling vectors $v$.

# 3 Expected Co-coercivity and Connection to Other Assumptions

Before introducing the condition of expected co-coercivity, we first review some details on co-coercivity, an intermediate notion between monotonicity and strong monotonicity [Zhu and Marcotte, 1996], and explain where it belongs as assumption in the literature of variational inequalities and min-max optimization.

**Co-coercive operators.** The co-coercive condition is relatively standard in operator splitting literature [Davis and Yin, 2017, Vũ, 2013] and for variational inequalities [Zhu and Marcotte, 1996]. It was used to analyze the celebrated forward-backward algorithm (a.k.a, proximal gradient) [Lions and Mercier, 1979, Chen and Rockafellar, 1997, Palaniappan and Bach, 2016] that is known not to converge for general monotone operators [Bauschke et al., 2011].

**Definition 3.1** (Co-coercivity / Co-coercive *around* $w^*$). We say that an operator $\xi$ is $\ell$–co-coercive if there exist $\ell > 0$ such that,[a] $\|\xi(x) - \xi(y)\|^2 \leq \ell \langle \xi(x) - \xi(y), x - y \rangle \quad \forall x, y \in \mathbb{R}^d$.
If there exist $w^* \in \mathbb{R}^d$ and $\ell > 0$ such that $\|\xi(x) - \xi(w^*)\|^2 \leq \ell \langle \xi(x) - \xi(w^*), x - w^* \rangle \quad \forall x \in \mathbb{R}^d$.
then we say that the operator $\xi$ is $\ell$–co-coercive *around* $w^*$. Note that in the last definition the point $w^*$ is not necessarily a point where $\xi(w^*) = 0$.

Note that from Cauchy-Swartz's inequality, one can get that a $\ell$-co-coercive operator is $\ell$-Lipschitz. In single-objective minization, one can show the converse statement by using convex duality. Thus, a gradient of a function is $L$–co-coercive if and only if the function is convex and $L$-smooth (i.e. $L$-Lipschitz gradients) [Bauschke et al., 2011]. However, in general, a $L$-Lipschitz operator is *not* $L$–co-coercive. What we can show instead is that a $L$-Lipschitz and $\mu$-strongly monotone operator is $\ell$–co-coercive with $\ell \in [L, L^2/\mu]$ [Facchinei and Pang, 2007]. Note that both ranges of the spectrum may occur. For instance, Chavdarova et al. [2019] present a suffcient condition in zero-sum games to have $\ell = O(L)$. Note that one can easily show that a sum of co-coercive operators is also co-coercive. Let us now provide a proposition summarizing the implications between (strong) monotonicity and co-coercivity.

**Proposition 3.2.** For a $L$-Lipschitz operator $\xi$, the following implications hold:

$$
\begin{array}{ccccc}
\mu\text{-strongly monotone} & \implies & \frac{L^2}{\mu}\text{-co-coercive} & \implies & \text{monotone} \\
\Downarrow & & \Downarrow & & \Downarrow \\
\mu\text{-quasi-strongly monotone} & \implies & \frac{L^2}{\mu}\text{-co-coercive around } x^* & \implies & \begin{array}{c}\text{variational} \\ \text{stability condition}\end{array}
\end{array}
$$

Let us also note that while a $\ell$-co-coercive operator is always $\ell$-Lipschitz continuous, it is possible for an operator to be $\ell$-co-coercive *around* $x^*$ and *not* be Lipschitz continuous. This highlights the wider applicability of the $\ell$-co-coercivity around $x^*$ assumption that is all we need for several of our convergence results, in contrast to the Lipschitz continuity of $\xi$ which is typically assumed in the variational inequality literature. In Appendix A.6, we provide such example of a $\mu$-quasi strongly monotone operator that is $\ell$-co-coercive around $x^*$, which is not monotone nor Lipschitz continuous.

## 3.1 Expected Co-coercivity (EC)

In our analysis of SGDA and SCO, we rely on a generic and remarkably weak assumption that we call *expected co-coercivity* (EC). In this section, we formally define EC, provide sufficient conditions for it to hold and relate it to the existing gradient assumptions.

**Assumption 3.3** (Expected Cocoercivity). We say that $\xi$ is $\ell_\xi$–co-coercive in expectation with respect to a distribution $\mathcal{D}$ if there exists $\ell_\xi > 0$ such that

$$
\mathbb{E}_{\mathcal{D}}\left[\|\xi_v(x) - \xi_v(x^*)\|^2\right] \leq \ell_\xi \langle \xi(x), x - x^* \rangle \quad \forall x \in \mathbb{R}^d. \tag{EC}
$$

For simplicity, we will write $\xi \in EC(\ell_\xi)$ to say that EC holds and we will refer to $\ell_\xi$ as the expected co-coercivity constant.

---

[a]Note that in our definition we consider the inverse of the co-coercive constant from Lions and Mercier [1979] which is the constant $\ell$ such that $\langle \xi(x) - \xi(y), x - y \rangle \geq \ell \|\xi(x) - \xi(y)\|^2$.

The convergence results in this paper will depend on the following operator noise at $x^*$ that is finite for any reasonable sampling distribution $\mathcal{D}$ for the sampling vector $v$:

$$\sigma^2 := \mathbb{E}_{\mathcal{D}}[\|\xi_v(x^*)\|^2] < \infty. \tag{6}$$

As we discuss below, common assumptions used to prove convergence of stochastic algorithms for solving the VI problem is uniform boundedness of the stochastic operator $\mathbb{E}\|\xi_i(x)\|^2 \leq c$ or uniform boundedness of the variance $\mathbb{E}\|\xi_i(x) - \xi(x)\|^2 \leq c$. However these assumptions either do not hold or are true only for restrictive set of problems. In our work we do not assume such bounds. Instead we use the following direct consequence of Assumption 3.3.

**Lemma 3.4.** If $\xi \in EC(\ell_\xi)$, then $\mathbb{E}\|\xi_v(x)\|^2 \leq 2\ell_\xi\langle\xi(x), x - x^*\rangle + 2\sigma^2$.

Let us now provide some more familiar sufficient conditions which guarantee that the EC condition holds and give closed form expression for the expected co-coercivity parameter.

**Proposition 3.5.** Let $\xi_i$ be $\ell_i$ co-coercive (or $\ell_i$ co-coercive around $x^*$), then $\xi \in EC(\ell_\xi)$. Let $\ell_{\max} = \max\{\ell_i\}_{i=1}^n$ and $\ell$ be the co-coercive constant of $\xi$, if we let $v$ to be a $b$-minibatch sampling, then $\ell_\xi = \frac{n}{b}\frac{b-1}{n-1}\ell + \frac{1}{b}\frac{n-b}{n-1}\ell_{\max}$ and $\sigma^2 = \frac{1}{b}\frac{n-b}{n-1}\sigma_1^2$, where $\sigma_1^2 := \frac{1}{n}\sum_{i=1}^n\|\xi_i(x^*)\|^2$.

In the above Proposition 3.5, we show how co-coercivity of $\xi_i$ implies expected co-coercivity. However, the opposite implication does not necessarily hold. Indeed the expected co-coercivity can hold even when we do not assume that $\xi_i$ are co-coercive, as we show in the next proposition.

**Proposition 3.6.** Let $\xi$ be quasi-strongly monotone and let $\xi_i$ be $L_i$-Lipschitz continuous for all $i \in [n]$. Then $\xi \in EC(\ell_\xi)$.

**Connection to Other Assumptions** In the optimization literature, the standard convergence analysis of stochastic gradient algorithms like SGD relied on bounded gradient ($\mathbb{E}\|\nabla f_i(x)\|^2 \leq c$) or bounded variance assumptions ($\mathbb{E}\|\nabla f_i(x) - \nabla f(x)\|^2 \leq c$) [Recht et al., 2011, Hazan and Kale, 2014, Rakhlin et al., 2012] or growth condition ($\mathbb{E}\|\nabla f_i(x)\|^2 \leq c_1\|\nabla f(x)\|^2 + c_2$) [Bottou et al., 2018, Schmidt et al., 2017]. However, a recent line of work shows that these assumptions might be restrictive or never be satisfied[5] and proposed alternative conditions [Nguyen et al., 2018, Vaswani et al., 2018, Gower et al., 2019, 2021, Khaled et al., 2020, Khaled and Richtárik, 2020, Assran et al., 2019, Koloskova et al., 2020, Patel and Zhang, 2021, Loizou et al., 2020, 2021]. One of the weakest assumptions used for the convergence analysis of SGD in the smooth setting, is expected smoothness (ES) proposed in Gower et al. [2019] (see last row of Table 1). Our expected co-coercivity condition (EC) can be seen as the generalization of ES in the operator setting.

In the literature of stochastic methods for solving the variational inequality problem and min-max optimization problem, similar assumptions have been made. In particular for the analysis of stochastic algorithms, papers assume either bounded operators [Nemirovski et al., 2009, Abernethy et al., 2021] or bounded variance [Juditsky et al., 2011, Yang et al., 2020, Lin et al., 2020a, Luo et al., 2020, Tran Dinh et al., 2020] and growth condition [Lin et al., 2020b]. In all of the these conditions, the values of parameters $c$, $c_1$ and $c_2$ (see Table 1) usually do not have a closed form expression – they are simply assumed to exist. However, to the best of our knowledge, there is no analysis using a concept similar to our expected co-coercivity. All existing analyses of SGDA for (quasi)-strongly monotone and co-coercive operators require the much stronger extra assumptions of "bounded noise" or "bounded variance" to guarantee convergence, while for SCO, there are no known convergence guarantees in the literature. Note that through Lemma 3.4, (EC) implies bounds on the gradient with closed-form problem-depended expressions for these constants. We also mention that other appropriate relaxed assumptions have been considered to prove the convergence of algorithms for (1) [Hsieh et al., 2020, Mishchenko et al., 2020, Chavdarova et al., 2019, Loizou et al., 2020], but not yet for SGDA nor SCO. For a wider literature review in the area, see Appendix E.

In Table 1, we illustrate the correspondence between conditions used in the stochastic optimization literature and the stochastic VI problem. For further connections between these conditions, see Appendix A.5. There, for example, we show why assuming bounded gradients together with strong monotonicity lead to an empty set of operators and explain why ES and EC (see last row of Table 1) are equivalent for convex and smooth single-objective optimization problems.

---

[5]For example, the bounded gradient assumption and strong convexity contradict each other in the unconstrained setting (see [Nguyen et al., 2018] for more details).

Table 1: Correspondence of Assumptions between Optimization and Variational Inequalities

| Assumptions | Stochastic Optimization $\min_x f(x) = \frac{1}{n}\sum_{i=1}^n f_i(x)$ | Stochastic Variational Inequality Find $x^*$ such that $\xi(x^*) = \frac{1}{n}\sum_{i=1}^n \xi_i(x^*) = 0$ |
|---|---|---|
| Bounded Gradient | $\mathbb{E}\|\nabla f_i(x)\|^2 \le c$ | $\mathbb{E}\|\xi_i(x)\|^2 \le c$ |
| Bounded Variance | $\mathbb{E}\|\nabla f_i(x) - \nabla f(x)\|^2 \le c$ | $\mathbb{E}\|\xi_i(x) - \xi(x)\|^2 \le c$ |
| Growth Condition | $\mathbb{E}\|\nabla f_i(x)\|^2 \le c_1\|\nabla f(x)\|^2 + c_2$ | $\mathbb{E}\|\xi_i(x)\|^2 \le c_1\|\xi(x)\|^2 + c_2$ |
| Expected Smoothness (ES) / Expected Cocoercivity (EC) | $\mathbb{E}\left[\|\nabla f_v(x) - \nabla f_v(x^*)\|^2\right] \le 2\mathcal{L}(f(x) - f(x^*))$ | $\mathbb{E}\left[\|\xi_v(x) - \xi_v(x^*)\|^2\right] \le \ell_\xi \langle \xi(x), x - x^* \rangle$ |

## 4 Stochastic Gradient Descent-Ascent

Having presented the update rule of SGDA (5) for solving the stochastic reformulation (4) of the original unconstrained stochastic variational inequality problem (1), let us now provide theorems for its convergence guarantees. We highlight that our theorems hold for any selection of distributions $\mathcal{D}$ over the random sampling vectors $v$ and as such they are able to describe the convergence of an infinite array of variants of SGDA each of which is associated with a specific probability law governing the data selection rule used to form minibatches.

**Theorem 4.1** (Constant Step-size). Assume that $\xi$ is $\mu$−quasi strongly monotone and that $\xi \in EC(\ell_\xi)$. Choose $\alpha_k = \alpha \le \frac{1}{2\ell_\xi}$ for all k. Then, the iterates of SGDA, given by (5), satisfy:

$$\mathbb{E}\left[\|x^k - x^*\|^2\right] \le (1 - \alpha\mu)^k \|x^0 - x^*\|^2 + \frac{2\alpha\sigma^2}{\mu}, \tag{7}$$

Note that we do not assume that $\xi$ or $\xi_i$ are monotone operators in Theorem 4.1. SGDA converges by only assuming that $\xi$ is quasi-strongly monotone and that EC holds. Theorem 4.1 states that SGDA converges linearly to a neighborhood of $x^*$ which is proportional to the step-size $\alpha$ and the noise at the optimum $\sigma^2$. We highlight, that since we control distribution $\mathcal{D}$ we also control the values of $\ell_\xi$ and $\sigma^2$, and in the case of $b$-minibatch sampling these values have a closed-form expressions as shown in Proposition 3.5. To the best of our knowledge, Theorem 4.1 is the first last-iterate non-asymptotic convergence guarantee for SGDA for solving quasi-strongly monotone problems without assuming extra conditions on the noise. It is worth mentioning that Lin et al. [2020b] also prove last-iterate convergence of SGDA for different class of problems (they do not assume quasi-strong monotonicity), but the proposed analysis requires much stronger noise conditions. In particular, a bound on the variance with vanishing constants is needed, which, as far as we know, can only be satisfied by running SGDA with growing mini-batch size [Friedlander and Schmidt, 2012] (see also App. E for a more detailed discussion). To highlight further the generality of Theorem 4.1, we note that for the deterministic GDA, $\sigma^2 = 0$. Thus, we can obtain the following corollary.

**Corollary 4.2** (Deterministic GDA). Let all assumptions of Theorem 4.1 be satisfied. Let $|S| = n$ with probability one (each iteration of SGDA uses a full batch gradient). Then by selecting $\alpha_k = \alpha \le \frac{1}{2\ell}$ for all k, the iterates of deterministic GDA satisfy: $\|x^k - x^*\|^2 \le (1 - \alpha\mu)^k \|x^0 - x^*\|^2$.

Even if Corollary 4.2 looks trivial, to the best of our knowledge, Theorem 4.1 is the first convergence theorem of SGDA that includes the deterministic GDA originally provided by Chen and Rockafellar [1997] as a special case.

**Optimal $b$-Minibatch Size:** Using standard computations, the convergence rate presented in Theorem 4.1 can be equivalently expressed as iteration complexity result as follows: If we are given any accuracy $\epsilon > 0$, choosing stepsize $\alpha = \min\left\{\frac{1}{2\ell_\xi}, \frac{\epsilon\mu}{4\sigma^2}\right\}$ and $k \ge \max\left\{\frac{2\ell_\xi}{\mu}, \frac{4\sigma^2}{\epsilon\mu^2}\right\} \log\left(\frac{2\|x^0 - x^*\|^2}{\epsilon}\right)$, implies $\mathbb{E}\|x^k - x^*\|^2 \le \epsilon$. By combining the lower bound on $k$ with the expressions of $\ell_\xi$ and $\sigma^2$ of Proposition 3.5, we have that the iteration complexity (by ignoring the logarithmic terms) becomes $k \ge \frac{2}{\mu}\max\{\ell_\xi, \frac{2\sigma^2}{\epsilon\mu}\}$ where $\ell_\xi = \frac{n}{b}\frac{b-1}{n-1}\ell + \frac{1}{b}\frac{n-b}{n-1}\ell_{\max}$ and $\sigma^2 = \frac{1}{b}\frac{n-b}{n-1}\sigma_1^2$. Thus, the total complexity of the algorithm as a function of the minibatch size $b$ is given by $TC(b) \le \frac{2}{\mu}\max\{b\ell_\xi, b\frac{2\sigma^2}{\epsilon\mu}\}$. By following the same steps with Gower et al. [2019], it can be shown that $b\ell_\xi$ is linearly increasing term in $b$ while $b\frac{2\sigma^2}{\epsilon\mu}$ is a linearly decreasing term in $b$. Hence, if we define $b^*$ to be the minibatch size that minimize the total complexity $TC(b)$ (optimal $b$-Minibatch Size) we have that if $\sigma_1^2 \le \ell_{\max}$ then $b^* = 1$ otherwise $b^* = n\frac{\ell - \ell_{\max} + \frac{2}{\epsilon\mu}\cdot\sigma_1^2}{n\ell - \ell_{\max} + \frac{2}{\epsilon\mu}\cdot\sigma_1^2}$.

In the next theorem, we provide an insightful stepsize-switching rule that describes when one should switch from a constant to a decreasing step-size regime to guarantee convergence to $x^*$ and not to a neighborhood, providing the first convergence analysis of SGDA under such a switching rule.

**Theorem 4.3.** Assume $\xi$ is $\mu$-quasi-strongly monotone and that $\xi \in EC(\ell_\xi)$. Let $\mathcal{K} := \ell_\xi/\mu$ and let $\alpha_k = \frac{1}{2\ell_\xi}$ for $k \le 4\lceil\mathcal{K}\rceil$ and $\alpha_k = \frac{2k+1}{(k+1)^2\mu}$ for $K > 4\lceil\mathcal{K}\rceil$. If $k \ge 4\lceil\mathcal{K}\rceil$, then iterates of SGDA, given by (5) satisfy:

$$\mathbb{E}\|x^k - x^*\|^2 \le \frac{\sigma^2}{\mu^2}\frac{8}{k} + \frac{16\lceil\mathcal{K}\rceil^2}{e^2 k^2}\|x^0 - x^*\|^2 = O\left(\frac{1}{k}\right). \tag{8}$$

## 5 Stochastic Consensus Optimization

The Consensus Optimization (CO) Algorithm is a computationally-light[6] second order methods which has been introduced in Mescheder et al. [2017] and it was shown to be an effective method for training GANs in a variety of settings. Liang and Stokes [2019] show first that CO converges linearly in the bilinear case. Abernethy et al. [2021] show that CO can be viewed as a perturbation of the deterministic Hamiltonian gradient descent (HGD) and explain how CO converges at the same rate as HGD, while Azizian et al. [2020] prove convergence of CO for $\mu$-strongly monotone operators with positive singular values of the Jacobian matrix $J = \nabla\xi$.

However, even if CO is used explicitly in the stochastic setting and in practice only minibatch variants are implemented, to the best of our knowledge all existing analysis focus only on the deterministic setting. Thus, our work is the first that provide convergence guarantees for the Stochastic Consensus Optimization (SCO) and due to our framework, our analysis includes the convergence of the deterministic update as special case. Impressively, our analysis provides tighter rates than previous analysis even in the deterministic setting.

For the results of this section, we assume that each $\xi_i$ in problem (1) is differentiable. That is, we have access to the Jacobian matrices $\mathbf{J}_i(x) = \nabla\xi_i(x)$. Following Loizou et al. [2020], in our analysis we will also assume that the Hamiltonian function is quasi-strongly convex and that it satisfies expected smoothness (see last row of Table 1). These are not strong assumptions, and as an example, they are satisfied for smooth bilinear min-max optimization problems Loizou et al. [2020]. In Appendix C, by extending the results of Loizou et al. [2020], we explain how these assumptions can be satisfied for the quadratic min-max problems.

### 5.1 Setting

The consensus optimization (CO) algorithm as presented in Mescheder et al. [2017] has the following update rule:

$$x^{k+1} = x^k - \alpha\xi(x^k) - \gamma\nabla\mathcal{H}(x^k) \tag{9}$$

where $\mathcal{H}(x) = \frac{1}{2}\|\xi(x)\|^2$ is the Hamiltonian function and $\alpha, \gamma > 0$ are the step-sizes. From its definition, it is clear that the update rule is essentially a weighted combination of GDA and the Hamiltonian gradient descent (HGD) of Balduzzi et al. [2018]. In practice, implementing CO in a mini-batch setting (stochastic) leads to biased estimates of the gradient of the Hamiltonian function [Mescheder et al., 2017]. This is one of the main reason that existing analysis was not able to capture the behavior of the method in the stochastic setting. However, recently Loizou et al. [2020] proposed a way to obtain unbiased estimators of the gradient of the Hamiltonian function by expressing the Hamiltonian of a stochastic game as a finite-sum problem. In this work, we adopt the finite-sum structure and the unbiased estimators proposed in Loizou et al. [2020] for the Hamiltonian part and we extend the formulation to capture the arbitrary sampling paradigm. That is, the Hamiltonian function can be expressed as, $\mathcal{H}(x) = \frac{1}{2}\|\xi(x)\|^2 = \frac{1}{n}\sum_{i=1}^{n}\frac{1}{n}\sum_{j=1}^{n}\frac{1}{2}\langle\xi_i(x), \xi_j(x)\rangle$. In addition, by following the stochastic reformulation setting presented in Section 2, let us have two independent random sampling vectors $u \sim \mathcal{D}$ and $v \sim \mathcal{D}$ and let us define: $\mathcal{H}_{u,v}(x) = \frac{1}{n}\sum_{i=1}^{n}\frac{1}{n}\sum_{j=1}^{n}\frac{1}{2}\langle u_i\xi_i(x), v_j\xi_j(x)\rangle$. Since vectors $u$ and $v$ are *independent* sampling vectors, it is clear that $\mathbb{E}_{u,v}[\mathcal{H}_{u,v}(x)] = \mathcal{H}(x)$, where $\mathbb{E}_{u,v}$ denotes the expectation with respect to distribution $\mathcal{D}$ on both vectors $u$ and $v$. Let us also

---

[6]At each step, CO requires only the computation of a Jacobian-vector product which can be efficiently evaluated in tasks like training neural networks with comparable computation time of a gradient [Pearlmutter, 1994].

use $\mathbf{J}_i(x) = \nabla\xi_i(x)$ to express the Jacobian matrices for $i$, then the gradient of $\mathcal{H}_{u,v}(x)$ has the following form:

$$\nabla\mathcal{H}_{u,v}(x) = \frac{1}{2}\left[\mathbf{J}_u^\top(x)\xi_v(x) + \mathbf{J}_v^\top(x)\xi_u(x)\right] \tag{10}$$

where $\mathbf{J}_u(x) = \frac{1}{n}\sum_{i=1}^n u_i\mathbf{J}_i(x)$ and $\mathbf{J}_v(x) = \frac{1}{n}\sum_{i=1}^n v_i\mathbf{J}_i(x)$. Similar to Loizou et al. [2020], it can be shown that $\nabla\mathcal{H}_{u,v}(x)$ is an unbiased estimator of $\nabla\mathcal{H}(x) = \mathbf{J}^\top(x)\xi(x)$. That is, $\mathbb{E}_{u,v}[\nabla\mathcal{H}_{u,v}(x)] = \nabla\mathcal{H}(x)$. Throughout this section, we will denote the gradient noise of the stochastic Hamiltonian function with $\sigma_\mathcal{H}^2 := \mathbb{E}_{u,v}[\|\nabla\mathcal{H}_{u,v}(x^*)\|^2]$, which is finite for any reasonable sampling distribution $\mathcal{D}$.

## 5.2 Stochastic Consensus Optimization and its Special Cases

Having explained the basic setting and background on consensus optimization algorithm and the Hamiltonian function, let us now present as Algorithm 1 the proposed stochastic consensus optimization (SCO) algorithm. Note that in each iteration $k$, two random sampling vectors $u$ and $v$ are sampled independently from a user-defined distribution $\mathcal{D}$. These vectors are used to evaluate $\xi_{v^k}$ and $\nabla\mathcal{H}_{v^k,u^k}(x^k)$, the unbiased estimators of $\xi(x^k)$ and $\nabla\mathcal{H}(x^k)$ at point $x^k$ respectively. Note also that the update rule of SCO

---

**Algorithm 1** Stochastic Consensus Opt. (SCO)

**Input:** Starting step-size $\alpha_0, \gamma_0 > 0$. Choose initial point $x^0 \in \mathbb{R}^d$. Distribution $\mathcal{D}$ of samples.
**for** $k = 0, 1, 2, \cdots, K$ **do**
    Sample independently $v^k \sim \mathcal{D}$ and $u^k \sim \mathcal{D}$
    Set step-sizes $\alpha_k$ and $\gamma_k$ according to a preselected step-size rule
    Set $x^{k+1} = x^k - \alpha_k\xi_{v^k}(x^k) - \gamma_k\nabla\mathcal{H}_{v^k,u^k}(x^k)$
**end for**
**Output:** The last iterate $x^k$

---

is a weighted combination of SGDA (5) and the stochastic Hamiltonian gradient descent (SHGD) of Loizou et al. [2020]. Thus, it is clear, that if one selects $\alpha^k = 0, \forall k > 0$ then the method is equivalent to SHGD and if $\gamma^k = 0, \forall k > 0$ then the method becomes equivalent to the SGDA. In addition if we select sampling vectors $u = v = (1, 1, \ldots, 1) \in R^n$ with probability 1, then from the definition of $\xi_{v^k}$ and $\nabla\mathcal{H}_{v^k,u^k}(x^k)$ we obtain the deterministic CO (9) as special case of our update rule.

## 5.3 Convergence Analysis

Let us now present our main theoretical results describing the performance of SCO. Similar to the previous section, we provide two main theorems for two different step-size selection.

**Theorem 5.1** (Constant Step-size). Assume $\xi$ is $\mu$-quasi-strongly monotone with $\mu \geq 0$ and that $\xi \in EC(\ell_\xi)$. Assume that the Hamiltonian function $\mathcal{H}$ is $\mu_\mathcal{H}$-quasi strongly convex and $\mathcal{L}_\mathcal{H}$-expected smooth. Then, for $\gamma_k = \gamma \leq \frac{1}{4\mathcal{L}_\mathcal{H}}$ and $\alpha_k = \alpha \leq \frac{1}{4\ell_\xi}$, the iterates of SCO satisfy:

$$\mathbb{E}\left[\|x^k - x^*\|^2\right] \leq (1 - \gamma\mu_\mathcal{H} - \alpha\mu)^k\|x^0 - x^*\|^2 + \frac{4[\alpha^2\sigma^2 + \gamma^2\sigma_\mathcal{H}^2]}{\gamma\mu_\mathcal{H} + \alpha\mu}. \tag{11}$$

If $\mu = 0$, that is $\xi$ only satisfies the variational stability condition $\langle\xi(x), x - x^*\rangle \geq 0$, then

$$\mathbb{E}\left[\|x^k - x^*\|^2\right] \leq (1 - \gamma\mu_\mathcal{H})^k\|x^0 - x^*\|^2 + \frac{4[\alpha^2\sigma^2 + \gamma^2\sigma_\mathcal{H}^2]}{\gamma\mu_\mathcal{H}}.$$

Theorem 5.1 is quite informative, as it highlights that both SGDA and SHG parts should coexist in the update rule of the SCO to guarantee faster convergence for $\mu$-quasi-strongly monotone operators (i.e. both step-sizes $\alpha$ and $\gamma$ should be positive up to specific values). However, if $\xi$ simply satisfies the variational stability condition $\langle\xi(x), x - x^*\rangle \geq 0$ (when $\mu = 0$), then the convergence rate of SCO, does not depend on the step-size $\alpha$ (the SGDA part) and the neighborhood of convergence $\frac{4[\alpha^2\sigma^2 + \gamma^2\sigma_\mathcal{H}^2]}{\gamma\mu}$ is smaller when $\alpha = 0$. Thus, in this case one needs to simply run the SHGD.

To appreciate the generality of Theorem 5.1, let us present some corollaries and compare the rates with existing results in the literature. First, we get a rate for the deterministic CO algorithm.

**Corollary 5.2** (Deterministic CO). Let all assumptions of Theorem 5.1 be satisfied. Then, for $\gamma_k = \gamma \leq \frac{1}{4L_\mathcal{H}}$ and $\alpha_k = \alpha \leq \frac{1}{4\ell}$, the iterates of CO satisfy: $\|x^k - x^*\|^2 \leq (1 - \gamma\mu_\mathcal{H} - \alpha\mu)^k \|x^0 - x^*\|^2$.

The result of Corollary 5.2 should be compared to the convergence guarantees for CO as provided in Abernethy et al. [2021], where the authors viewed CO as a perturbation of the deterministic Hamiltonian gradient descent (HGD). In particular, Abernethy et al. [2021] gives the rate $1 - \frac{\mu_\mathcal{H}}{4L_\mathcal{H}}$ for CO under similar assumptions,[7] which is clearly slower than our $1 - \frac{\mu_\mathcal{H}}{4L_\mathcal{H}} - \frac{\mu}{4\ell}$ rate. The authors had explicitly noted that treating the GDA part as an adversarial perturbation most likely should be improved upon. With our analysis, we provide a different, more natural analysis of CO and answer this open problem. In addition, note that by setting $\gamma_k = 0$ and $\alpha_k = \alpha < 1/2\ell_\xi$, then SCO becomes equivalent to SGDA and Theorem 5.1 matches the convergence guarantees presented in Theorem 4.1. On the other hand, by setting $\alpha^k = 0$, SCO yields equivalent updates to SHGD and our result matches the theoretical guarantees of Loizou et al. [2020] as we show in the next corollary.

**Corollary 5.3.** Under the assumptions of Thm. 5.1, set $\alpha^k = 0$ and $\gamma_k = \gamma \leq 1/2\mathcal{L}_\mathcal{H}$. Then SCO is equivalent to SHGD and its iterates satisfy: $\mathbb{E}\left[\|x^k - x^*\|^2\right] \leq (1 - \gamma\mu_\mathcal{H})^k \|x^0 - x^*\|^2 + \frac{2\gamma\sigma_\mathcal{H}^2}{\mu_\mathcal{H}}$.

All previous results for SCO show convergence to a neighborhood of $x^*$. In the next theorem, by selecting decreasing step-sizes (switching strategy) for the values of $\alpha$ and $\gamma$, we are able to guarantee a sublinear convergence to the exact solution for SCO. To the best of our knowledge this is the first result analyzing SCO with decreasing step-sizes.

**Theorem 5.4.** Assume $\xi$ is $\mu$-quasi-strongly monotone and that $\xi \in EC(\ell_\xi)$. Assume that the Hamiltonian function $\mathcal{H}$ is $\mu_\mathcal{H}$-quasi strongly convex and $\mathcal{L}_\mathcal{H}$-expected smooth. Let $\alpha_k = \gamma_k$, $\psi = \max\{\ell_\xi, \mathcal{L}_\mathcal{H}\}$ and $k^* := 8\frac{\psi}{\mu_\mathcal{H}+\mu}$. Let also, $\gamma_k = \frac{1}{4\psi}$ for $k \leq \lceil k^* \rceil$ and $\gamma_k = \frac{2k+1}{(k+1)^2[\mu_\mathcal{H}+\mu]}$ for $k > \lceil k^* \rceil$. If $k \geq \lceil k^* \rceil$, then SCO iterates satisfy:

$$\mathbb{E}\|x^k - x^*\|^2 \leq \frac{\sigma_\mathcal{H}^2 + \sigma^2}{[\mu + \mu_\mathcal{H}]^2}\frac{16}{k} + \frac{(k^*)^2}{e^2 k^2}\|x^0 - x^*\|^2 = O\left(\frac{1}{k}\right) \tag{12}$$

If $\mu = 0$, that is $\xi$ only satisfies the variational stability condition $\langle \xi(x), x - x^* \rangle \geq 0$, then SCO is still able to converge sublinearly with $O\left(\frac{1}{k}\right)$, to $x^*$.

# 6 Numerical Evaluation

The purpose of this experimental section is to corroborate our theoretical results, which form the main contributions of this paper. To do so, we focus on strongly-monotone quadratic games of the following form:

$$\min_{x_1 \in \mathbb{R}^d} \max_{x_2 \in \mathbb{R}^p} \frac{1}{n}\sum_i \frac{1}{2}x_1^\top \mathbf{A}_i x_1 + x_1^\top \mathbf{B}_i x_2 - \frac{1}{2}x_2^\top \mathbf{C}_i x_2 + a_i^\top x_1 - c_i^\top x_2 \tag{13}$$

We show that the Hamiltonian function of such a game is $\mu_\mathcal{H}$-quasi-strongly convex and $L_\mathcal{H}$-smooth in App. C.1. For the game to also be strongly-monotone and co-coercive, we sample the matrices such that $\mu_A \mathbf{I} \preceq \mathbf{A}_i \preceq L_A \mathbf{I}$, $\mu_C \mathbf{I} \preceq \mathbf{C}_i \preceq L_C \mathbf{I}$, and $\mu_B^2 \mathbf{I} \preceq \mathbf{B}_i^\top \mathbf{B}_i \preceq L_B^2 \mathbf{I}$, where $\mathbf{I}$ is the identity matrix; the exact sampling is described in App. C.2. The bias terms $a_i, c_i$ are sampled from a normal distribution. We pick the step-size for the different methods according to our theoretical findings. That is, for constant step-size, we select $\alpha = \frac{1}{2\ell_\xi}$ for SGDA (Theorem 4.1), $\alpha = \frac{1}{4\ell_\xi}, \gamma = \frac{1}{4\mathcal{L}_\mathcal{H}}$ for SCO (Theorem 5.1), and $\gamma = \frac{1}{2\mathcal{L}_\mathcal{H}}$ for SHGD (Corollary 5.3). For the stepsize-switching rule that guarantees convergence to $x^*$, we use the step-sizes proposed in Theorem 4.3 for SGDA and Theorem 5.4 for SCO. Let us first look at the qualitative behavior of SGDA, SCO and SHGD on a simple 2d example where $x_1, x_2 \in \mathbb{R}$. We show the trajectories of the different algorithms in Fig. 1a. As expected we observe that the behavior of SCO is in between SGDA and SHGD. Recall that the update rule of SCO is a weighted combination of SGDA and the SHGD. The code to reproduce our results can be found at https://github.com/hugobb/StochasticGamesOpt.

**Comparison of Algorithms (Constant step-size).** We look at the convergence of the methods for different games where we vary the condition number $\kappa_G = \frac{\ell_\xi}{\mu}$. In comparing the methods, we

---

[7]For this convergence rate, Abernethy et al. [2021] assumed that the Hamiltonian function satisfies the Polyak-Lojasiewicz condition but focus on strongly-convex and strongly-concave min-max optimization problems.

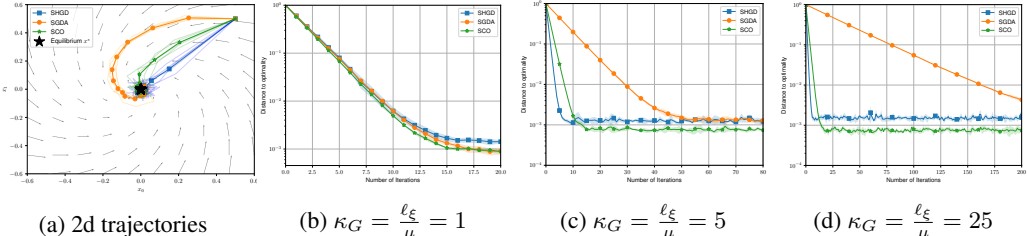

| (a) 2d trajectories | (b) $\kappa_G = \frac{\ell_\xi}{\mu} = 1$ | (c) $\kappa_G = \frac{\ell_\xi}{\mu} = 5$ | (d) $\kappa_G = \frac{\ell_\xi}{\mu} = 25$ |

Figure 1: Convergence of SGDA, SCO and SHGD on different quadratic games. **(a)** Trajectories of SCO, SHGD and SGDA on a 2d quadratic game. The arrows represent the direction defined by $\xi(x)$ at a particular point $x$. **(b-d)** Distance to optimality $\frac{\|x^k - x^*\|^2}{\|x^0 - x^*\|^2}$ as a function of the number of iterations. Each plot corresponds to a game with a particular condition number $\kappa_G = \frac{\ell_\xi}{\mu}$. The solid lines represent the average performance over the 5 runs and the colored area represent the 95% confidence intervals.

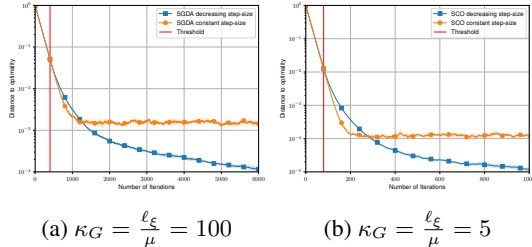

| (a) $\kappa_G = \frac{\ell_\xi}{\mu} = 100$ | (b) $\kappa_G = \frac{\ell_\xi}{\mu} = 5$ |

Figure 2: Comparison between constant and decreasing step size regimes of SGDA and SCO. The vertical red lines correspond to the moment we switch from a constant to a decreasing step-size. The solid lines represent the average performance over the 5 runs and the colored area represent the 95% confidence intervals.

use the relative distance to optimality $\frac{\|x^k - x^*\|^2}{\|x^0 - x^*\|^2}$. As predicted from our theoretical results, when constant step-size is used, all the methods converge linearly to a neighborhood of the solution (see Figure 1). We observe that the performance of SGDA depends on the condition number: the higher the condition number, the slower the convergence. In contrast, the convergence of both SHGD and SCO is less affected by a larger condition number. We also observe that SGDA is slower than SHGD, but converges to a smaller neighborhood of the solution (see e.g. Fig. 1b). SCO achieves a good trade-off; it converges fast like SHGD and to a small neighborhood like SGDA. An important note is that the size of the neighborhood heavily depends on the selection of the learning rate; we explore this dependence in App. C.3.

**Constant vs Decreasing step-size.** We also compare the performance of SGDA and SCO in the constant and decreasing step-size regimes considered in Theorems 4.1 and 4.3 for SGDA and Theorems 5.1 and 5.4 for SCO. We present our results in Figure 2. As predicted from our theoretical analysis for both methods, the decreasing step-size (switching step-size rule) reaches higher precision compare to the constant step-size. In Figure 2a the vertical red line denotes the value $4\lceil \ell_\xi / \mu \rceil$ predicted in Theorem 4.3 while in Figure 2b the red line denotes the value $\left\lceil 8\frac{\max\{\ell_\xi, \mathcal{L}_\mathcal{H}\}}{\mu_\mathcal{H} + \mu}\right\rceil$ predicted in Theorem 5.4. Note that for both algorithms the red line is a good approximation of the point where SGDA and SCO need to change their update rules from constant to decreasing step-size.

# 7 Conclusion and Future Directions of Research

We provided the first last-iterate convergence analysis of SGDA and SCO without requiring any strong bounded noise assumption, by introducing the much weaker expected co-coercivity assumption. We proved last-iterate convergence for both methods for a class of unconstrained variational inequality problems that are potentially non-monotone (quasi-strongly monotone problems), with both constant and decreasing step-sizes. Future work includes extending our results beyond the $\mu$-quasi strongly monotone assumption for SGDA (assuming only co-coercivity), where we expect to obtain a slower sublinear rate. We also believe the proposal of expected co-coercivity to be of independent interest; it could be used to provide an efficient analysis of other algorithms to solve (1) under the arbitrary sampling paradigm, and it would also be interesting to generalize the analysis to the constrained formulation of variational inequalities.

## Acknowledgments and Disclosure of Funding

**Funding.** Nicolas Loizou acknowledges support by the IVADO Postdoctoral Funding Program. This research was partially supported by the Canada CIFAR AI Chair Program and by a Google Focused Research award. Simon Lacoste-Julien is a CIFAR Associate Fellow in the Learning in Machines & Brains program.

**Competing interests.** Simon Lacoste-Julien additionally works part time as the head of the SAIT AI Lab, Montreal from Samsung. Hugo Berard was working part time as a research intern at Facebook, Montreal.

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
