# Supplementary Material

The supplementary material is organized as follows: In Section A, we give some basic definitions and provide the proofs of the propositions, lemmas and theorems related to the expected co-coercivity condition as presented in Section 3 of the main paper. In Section B we present the proofs of the main theorems and corollaries for the convergence of SGDA and SCO. In Section C we present the experimental details and provide additional experiments. Finally in Section D we explain how our convergence results can be easily adapted to the general stochastic setting and in Section E we provide further related work.

## Contents

## A Proofs of Results on Co-coercivity and Expected Co-coercivity

Let us start by re-stating the main definitions of the classes of operators under study.

**Definition A.1** (Lipschitz continuous). An operator $\xi : \mathbb{R}^d \to \mathbb{R}^d$ is $L-$Lipschitz continuous if there is $L > 0$ such that:

$$\|\xi(x) - \xi(y)\| \leq L\|x - y\|, \quad \forall x, y \in \mathbb{R}^d \tag{14}$$

**Definition A.2** (Co-coercivity). We say that an operator $\xi$ is $\ell$–co-coercive if there exist $\ell > 0$ such that:

$$\|\xi(x) - \xi(y)\|^2 \leq \ell\langle\xi(x) - \xi(y), x - y\rangle \quad \forall x, y \in \mathbb{R}^d.$$

**Definition A.3** (Co-coercive *around* $w^*$). We say that an operator $\xi$ is $\ell$–co-coercive *around* $w^*$ if there exist $w^* \in \mathbb{R}^d$ and $\ell > 0$ such that

$$\|\xi(x) - \xi(w^*)\|^2 \leq \ell\langle\xi(x) - \xi(w^*), x - w^*\rangle \quad \forall x \in \mathbb{R}^d.$$

Note that in this definition, the point $w^*$ is not necessarily a point where $\xi(w^*) = 0$.

**Definition A.4** (Strongly monotone / monotone). We say that an operator $\xi$ is $\mu$–strongly monotone if there exist $\mu > 0$ such that

$$\langle\xi(x) - \xi(y), x - y\rangle \geq \mu\|x - y\|^2 \quad \forall x, y \in \mathbb{R}^d.$$

If $\mu = 0$, that is

$$\langle\xi(x) - \xi(y), x - y\rangle \geq 0 \quad \forall x, y \in \mathbb{R}^d,$$

then we say that the operator is monotone.

**Definition A.5** (Quasi-Strongly Monotone / Variational Stability Condition). We say that an operator $\xi$ is $\mu$-*quasi-strongly monotone* if there exist $\mu > 0$ such that

$$\langle\xi(x), x - x^*\rangle \geq \mu\|x - x^*\|^2 \quad \forall x \in \mathbb{R}^d.$$

Here $x^*$ is the solution of the stochastic variational inequality problem (1). If $\mu = 0$, that is

$$\langle\xi(x), x - x^*\rangle \geq 0 \tag{15}$$

then we say that $\xi$ satisfies the variational stability condition.

### A.1 Proof of Proposition 3.2

Before stating the proof of Proposition 3.2, we clarify that the assumption of $L$-Lipschitzness of $\xi$ is only used for the implications where $L$ appear; it is not needed for the other implications. In particular, while a $\ell$-co-coercive operator is always $\ell$-Lipschitz continuous by using Cauchy-Schwartz,[8] it is possible for an operator to be $\ell$-co-coercive *around* $x^*$ and *not* be Lipschitz continuous (see such an example in Section A.6). This highlights the wider applicability of the $\ell$-co-coercivity around $x^*$ assumption that is all we need for several of our convergence results, in contrast to the Lipschitz continuity of $\xi$ which is typically assumed in the variational inequality literature.

---

[8] $\|\xi(x) - \xi(x')\|^2 \leq \ell\langle\xi(x) - \xi(x'), x - x'\rangle \leq \ell\|\xi(x) - \xi(x')\|\|x - x'\| \implies \|\xi(x) - \xi(x')\| \leq \ell\|x - x'\|.$

*Proof.* Most of these implications can be found in Facchinei and Pang [2007].

$\mu$-strongly monotone $\implies \frac{L^2}{\mu}$-co-coercive: The proof of this result is a direct application of strong monotonicity and Lipschitzness properties:

$$\|\xi(x) - \xi(x')\|^2 \le L^2\|x - x'\|^2 \le \frac{L^2}{\mu}\langle \xi(x) - \xi(x'), x - x'\rangle, \qquad \forall x, x' \in \mathbb{R}^d.$$

$\ell$-co-coercive $\implies$ monotone: It comes from the fact that a norm is non-negative.

$$0 \le \|\xi(x) - \xi(x')\|^2 \le \ell\langle \xi(x) - \xi(x'), x - x'\rangle, \qquad \forall x, x' \in \mathbb{R}^d.$$

monotone $\implies$ variational stability condition: It comes from the fact that monotonicity applied to $x' = x^*$ is variational stability condition.

Quasi $\mu$-strongly monotone $\implies \frac{L^2}{\mu}$-co-coercive relatively to $x^*$: (It is the only implication that is not proven in Facchinei and Pang [2007]) The proof of this result is a direct application of quasi-strong monotonicity and Lipschitzness properties:

$$\|\xi(x) - \xi(x^*)\|^2 \le L^2\|x - x^*\|^2 \le \frac{L^2}{\mu}\langle \xi(x), x - x^*\rangle, \qquad \forall x \in \mathbb{R}^d.$$

$\frac{L^2}{\mu}$-co-coercive relatively to $x^* \implies$ variational stability condition: It comes from the fact that a norm is non-negative.

$$0 \le \|\xi(x) - \xi(x^*)\|^2 \le \ell\langle \xi(x), x - x^*\rangle, \qquad \forall x \in \mathbb{R}^d.$$

$\square$

### A.2 Proof of Lemma 3.4

*Proof.*
$$
\begin{aligned}
\mathbb{E}_{\mathcal{D}}\|\xi_v(x)\|^2 &= \mathbb{E}_{\mathcal{D}}\|\xi_v(x) - \xi_v(x^*) + \xi_v(x^*)\|^2 \\
&\le 2\mathbb{E}_{\mathcal{D}}\|\xi_v(x) - \xi_v(x^*)\|^2 + 2\mathbb{E}_{\mathcal{D}}\|\xi_v(x^*)\|^2 \\
&\overset{EC}{\le} 2\ell_\xi\langle \xi(x), x - x^*\rangle + 2\mathbb{E}_{\mathcal{D}}\|\xi_v(x^*)\|^2 \\
&\overset{(6)}{\le} 2\ell_\xi\langle \xi(x), x - x^*\rangle + 2\sigma^2.
\end{aligned}
\tag{16}
$$
The first inequality follows from the estimate $\|a + b\|^2 \le 2\|a\|^2 + 2\|b\|^2$. $\square$

### A.3 Proof of Proposition 3.5

Before we formally present the proof of Proposition 3.5, let us first establish some random set terminology.

Let $C \subseteq [n]$ and let $e_C := \sum_{i \in C} e_i$, where $\{e_1, \dots, e_n\}$ are the standard basis vectors in $\mathbb{R}^n$. These subsets will be selected using a random set valued map $S$, in the literature referred to by the name *sampling*. A sampling is uniquely characterized by choosing subset probabilities $p_C \ge 0$ for all subsets $C$ of $[n]$:

$$\mathbb{P}[S = C] = p_C, \quad \forall C \subset [n], \tag{17}$$

where $\sum_{C \subseteq [n]} p_C = 1$. In this work, following the terminology of Gower et al. [2019, 2021], our results hold for *proper* samplings.

**Definition A.6.** A sampling $S$ is called proper if $p_i \overset{\text{def}}{=} \mathbb{P}[i \in S] = \sum_{C : i \in C} p_C$ is positive for all $i$.

As we mentioned in the main paper, in this work we focus on $b$-minibatch sampling (see Definition 2.1) however we highlight again that our results hold for the larger class of sampling vectors $v \in \mathbb{R}^n$ that satisfy $\mathbb{E}_{\mathcal{D}}[v_i] = 1$, for $i = 1, \dots, n$.

For example, the random vector $v = v(S)$ given by $v = \sum_{i \in S} \frac{1}{p_i} e_i$ is a sampling vector. This can be easily proved, by noticing that $v_i = \mathbf{1}_{(i \in S)}/p_i$, where $\mathbf{1}_{(i \in S)}$ is the indicator function of the event

$i \in S$. Then, It follows that $\mathbb{E}[v_i] = \mathbb{E}\left[\mathbf{1}_{(i \in S)}\right]/p_i = 1$. Commonly used samplings that captured by our theory are the *independent sampling, partition sampling, single-element sampling and importance sampling*. For more details on these different samplings check Gower et al. [2019].

By definition 2.1 of $b$-minibatch sampling, it holds that $\mathbb{P}[i \in S] = p_i = \frac{b}{n}$, and $\mathbb{P}[i, j \in S] = \frac{b}{n}\frac{b-1}{n-1}$.

All the sampling schemes presented in Gower et al. [2019] had the following additional property: there exists a constant $z > 0$ such that

$$\frac{\mathbb{P}[i, j \in S]}{\mathbb{P}[i \in S]\,\mathbb{P}[j \in S]} = z, \quad \forall i, j \in \{1, \dots, n\},\ i \neq j. \tag{18}$$

For $b$-minibatch sampling, $z = \frac{n}{b}\frac{b-1}{n-1}$ [Gower et al., 2021].

Let us now present the proof of Proposition 3.5.

*Proof.* Since $\xi$ is $\ell_i$–co-coercive *around* $x^*$ then we have that $\xi$ is $\ell$–co-coercive *around* $x^*$. That is $\forall x \in \mathbb{R}^d$ it holds,

$$\|\xi_i(x) - \xi_i(x^*)\|^2 \le \ell_i \langle \xi_i(x) - \xi_i(x^*), x - x^* \rangle \tag{19}$$

$$\|\xi(x) - \xi(x^*)\|^2 \le \ell \langle \xi(x) - \xi(x^*), x - x^* \rangle = \ell \langle \xi(x), x - x^* \rangle. \tag{20}$$

Noticing that

$$\|\xi_v(x) - \xi_v(x^*)\|^2 = \frac{1}{n^2}\left\|\sum_{i \in S}\frac{1}{p_i}(\xi_i(x) - \xi_i(x^*))\right\|^2$$

$$= \sum_{i,j \in S}\left\langle \frac{1}{np_i}(\xi_i(x) - \xi_i(x^*)), \frac{1}{np_j}(\xi_j(x) - \xi_j(x^*))\right\rangle,$$

we have

$$\mathbb{E}[\|\xi_v(x) - \xi_v(x^*)\|^2] = \sum_C p_C \sum_{i,j \in C}\left\langle \frac{1}{np_i}(\xi_i(x) - \xi_i(x^*)), \frac{1}{np_j}(\xi_j(x) - \xi_j(x^*))\right\rangle$$

$$= \sum_{i,j=1}^n \sum_{C:i,j \in C} p_C \left\langle \frac{1}{np_i}(\xi_i(x) - \xi_i(x^*)), \frac{1}{np_j}(\xi_j(x) - \xi_j(x^*))\right\rangle$$

$$= \sum_{i,j=1}^n \frac{\mathbb{P}[i, j \in S]}{p_i p_j}\left\langle \frac{1}{n}(\xi_i(x) - \xi_i(x^*)), \frac{1}{n}(\xi_j(x) - \xi_j(x^*))\right\rangle,$$

where we used a double counting argument in the 2nd equality. Now since $\mathbb{P}[i, j \in S]/(p_i p_j) = z$ for $i \neq j$ (18) and $\mathbb{P}[i, i \in S] = p_i$ we have from the above that

$$\mathbb{E}[\|\xi_v(x) - \xi_v(x^*)\|^2] = \sum_{i \neq j} z \left\langle \frac{1}{n}(\xi_i(x) - \xi_i(x^*)), \frac{1}{n}(\xi_j(x) - \xi_j(x^*))\right\rangle$$

$$+ \sum_{i=1}^n \frac{1}{n^2}\frac{1}{p_i}\|\xi_i(x) - \xi_i(x^*))\|^2$$

$$= \sum_{i,j=1}^n z \left\langle \frac{1}{n}(\xi_i(x) - \xi_i(x^*)), \frac{1}{n}(\xi_j(x) - \xi_j(x^*))\right\rangle$$

$$+ \sum_{i=1}^n \frac{1}{n^2}\frac{1}{p_i}(1 - p_i z)\|\xi_i(x) - \xi_i(x^*))\|^2 \tag{21}$$

$$\overset{(19)}{\le} z \|\xi(x) - \xi(x^*)\|^2$$

$$+ \sum_{i=1}^n \frac{1}{n^2}\frac{\ell_i}{p_i}(1 - p_i z)\langle \xi_i(x) - \xi_i(x^*), x - x^* \rangle$$

$$\overset{(20)}{\le} \left(z\ell + \max_{i=1,\dots,n}\frac{\ell_i}{np_i}(1 - p_i z)\right)\langle \xi(x), x - x^* \rangle. \tag{22}$$

Comparing the above to the definition of expected co-coercivity (EC) we have that

$$\ell_\xi = z\ell + \max_{i=1,\dots,n} \frac{\ell_i}{np_i}(1 - p_i z).\tag{23}$$

Using that for $b$-minibatch sampling it holds $\mathbb{P}\left[i \in S\right] = p_i = \frac{b}{n}$ and $z = \frac{n}{b}\frac{b-1}{n-1}$ we obtain

$$\ell_\xi = \frac{n}{b}\frac{b-1}{n-1}\ell + \frac{1}{b}\frac{n-b}{n-1}\ell_{\max},$$

where $\ell_{\max} = \max\{\ell_i\}_{i=1}^n$.

The specialized expressions of $\sigma^2$ for the $b$-minibatch sampling, can be obtain by following the same steps of Proposition 3.10 of Gower et al. [2019]. Below using our notation, we include this derivation for completeness:

$$
\begin{aligned}
\sigma^2 = \mathbb{E}_{\mathcal{D}}[\|\xi_v(x^*)\|^2] &= \mathbb{E}\left[\left\|\frac{1}{n}\sum_{i=1}^n \nabla f_i(x^*)v_i\right\|^2\right] = \frac{1}{n^2}\mathbb{E}\left[\left\|\sum_{i=1}^n \nabla f_i(x^*)v_i\right\|^2\right] = \frac{1}{n^2}\mathbb{E}\left[\left\|\sum_{i\in S}\frac{1}{p_i}\xi_i(x^*)\right\|^2\right]\\
&= \frac{1}{n^2}\mathbb{E}\left[\left\|\sum_{i=1}^n 1_{i\in S}\frac{1}{p_i}\xi_i(x^*)\right\|^2\right] = \frac{1}{n^2}\mathbb{E}\left[\sum_{i=1}^n\sum_{j=1}^n 1_{i\in S}1_{j\in S}\langle\frac{1}{p_i}\xi_i(x^*),\frac{1}{p_j}\xi_j(x^*)\rangle\right]\\
&= \frac{1}{n^2}\sum_{i,j}\frac{\mathbb{P}\left[i,j\in S\right]}{p_i p_j}\langle\xi_i(x^*),\xi_j(x^*)\rangle.
\end{aligned}
$$

Recall that for $b$-minibatch sampling, $\mathbb{P}\left[i\in S\right] = p_i = \frac{b}{n}$, and $\mathbb{P}\left[i,j\in S\right] = \frac{b}{n}\frac{b-1}{n-1}$. Thus,

$$
\begin{aligned}
\sigma^2 &= \frac{1}{n^2}\sum_{i,j\in[n]}\frac{\mathbb{P}\left[i,j\in S\right]}{p_i p_j}\langle\xi_i(x^*),\xi_j(x^*)\rangle\\
&= \frac{1}{n^2}\sum_{i\neq j}\frac{b(b-1)}{n(n-1)}\cdot\frac{n^2}{b^2}\langle\xi_i(x^*),\xi_j(x^*)\rangle + \frac{1}{n^2}\sum_{i\in[n]}\frac{n}{b}\|\xi_i(x^*)\|^2\\
&= \frac{1}{nb}\left(\sum_{i\neq j}\frac{b-1}{n-1}\langle\xi_i(x^*),\xi_j(x^*)\rangle + \sum_{i\in[n]}\|\xi_i(x^*)\|^2\right)\\
&= \frac{1}{nb}\left(\sum_{i,j\in[n]}\frac{b-1}{n-1}\langle\xi_i(x^*),\xi_j(x^*)\rangle + \sum_{i\in[n]}\frac{n-b}{n-1}\|\xi_i(x^*)\|^2\right)\\
&= \frac{1}{nb}\cdot\frac{n-b}{n-1}\sum_{i\in[n]}\|\xi_i(x^*)\|^2\\
&= \frac{1}{b}\cdot\frac{n-b}{n-1}\sigma_1^2,
\end{aligned}
$$

where $\sigma_1^2 := \frac{1}{n}\sum_{i=1}^n\|\xi_i(x^*)\|^2$. $\qquad\square$

### A.4 Proof of Proposition 3.6

**Proposition A.7.** Let $\xi$ be $\mu$-quasi-strongly monotone and let $\xi_i$ be $L_i$-Lipschitz continuous for all $i \in [n]$. Then $\xi \in EC(\ell_\xi)$.

For a general proper sampling scheme (Def. A.6), we can provide the following (loose) bound on $\ell_\xi$:

$$\ell_\xi \leq \frac{1}{n}\sum_{i=1}^{n} \frac{\mathbb{E}_{\mathcal{D}}\left[v_i^2\right] L_i^2}{\mu}. \tag{24}$$

If, as is the case for standard sampling schemes from Gower et al. [2019], we assume that there exists a $z > 0$ such that $z = \frac{\mathbb{P}[i, j \in S]}{\mathbb{P}[i \in S]\mathbb{P}[j \in S]}$ for all $i, j \in \{1, \ldots, n\}$, $i \neq j$, then we can use the tighter value:

$$\ell_\xi = \left(zL^2 + \sum_{i=1}^{n} \frac{1}{n^2} \frac{L_i^2}{p_i}\left(1 - p_i z\right)\right)\frac{1}{\mu}, \tag{25}$$

where $L$ is the Lipschitz continuous parameter of operator $\xi$ and $p_i = \mathbb{P}\left[i \in S\right]$.

Finally, for $b$-minibatch sampling, it holds $p_i = \frac{b}{n}$ and $z = \frac{n}{b}\frac{b-1}{n-1}$ and thus $\xi \in EC(\ell_\xi)$ with

$$\ell_\xi = \left(\frac{n}{b}\frac{b-1}{n-1}L^2 + \left(\frac{1}{n}\sum_{i=1}^{n}L_i^2\right)\frac{1}{b}\frac{n-b}{n-1}\right)\frac{1}{\mu}. \tag{26}$$

*Proof.* Since $\xi_i$ is $L_i$–Lipschitz continuous for all $i$, then we have that $\xi$ is $L$–Lipschitz continuous (with $L \leq \frac{1}{n}\sum_i L_i$ by using Jensen's inequality on $\|\cdot\|^2$). That is, $\forall x \in \mathbb{R}^d$ it holds:

$$\|\xi_i(x) - \xi_i(y)\| \leq L_i\|x - y\|, \quad \forall x, y \in \mathbb{R}^d \tag{27}$$

$$\|\xi(x) - \xi(y)\| \leq L\|x - y\|, \quad \forall x, y \in \mathbb{R}^d \tag{28}$$

We first prove the general case:

$$
\begin{aligned}
\mathbb{E}_{\mathcal{D}}[\|\xi_v(x) - \xi_v(x^*)\|^2] &= \mathbb{E}_{\mathcal{D}}\left[\|\frac{1}{n}\sum_{i=1}^{n}v_i\xi_i(x) - \frac{1}{n}\sum_{i=1}^{n}v_i\xi_i(x^*)\|^2\right] \\
&\overset{\text{Jensen's}}{\leq} \mathbb{E}_{\mathcal{D}}\left[\frac{1}{n}\sum_{i=1}^{n}\|v_i[\xi_i(x) - \xi_i(x^*)]\|^2\right] \\
&= \mathbb{E}_{\mathcal{D}}\left[\frac{1}{n}\sum_{i=1}^{n}v_i^2\|\xi_i(x) - \xi_i(x^*)\|^2\right] \\
&\overset{(27)}{\leq} \mathbb{E}_{\mathcal{D}}\left[\frac{1}{n}\sum_{i=1}^{n}v_i^2 L_i^2\|x - x^*\|^2\right] \\
&= \frac{1}{n}\sum_{i=1}^{n}\mathbb{E}_{\mathcal{D}}\left[v_i^2\right]L_i^2\|x - x^*\|^2 \\
&\overset{(2)}{\leq} \frac{1}{n}\sum_{i=1}^{n}\frac{\mathbb{E}_{\mathcal{D}}\left[v_i^2\right]L_i^2}{\mu}\langle\xi(x), x - x^*\rangle,
\end{aligned}
$$

yielding (24).

The use of Jensen's inequality above is the source of looseness in the bound. With the $z$ constant property, we can avoid it with the following derivations

By following the same steps to the proof of Proposition 3.5, we obtain (21). That is,

$$
\begin{aligned}
\mathbb{E}[\|\xi_v(x) - \xi_v(x^*)\|^2] &= \sum_{i,j=1}^{n} z\left\langle\frac{1}{n}(\xi_i(x) - \xi_i(x^*)), \frac{1}{n}(\xi_j(x) - \xi_j(x^*))\right\rangle \\
&+ \sum_{i=1}^{n}\frac{1}{n^2}\frac{1}{p_i}\left(1 - p_i z\right)\|\xi_i(x) - \xi_i(x^*)\|^2.
\end{aligned}
$$

Now by using (27) and (28), we obtain the following

$$
\begin{aligned}
\mathbb{E}[\|\xi_v(x) - \xi_v(x^*)\|^2] &= \sum_{i,j=1}^{n} z \left\langle \frac{1}{n}(\xi_i(x) - \xi_i(x^*)), \frac{1}{n}(\xi_j(x) - \xi_j(x^*)) \right\rangle \\
&\quad + \sum_{i=1}^{n} \frac{1}{n^2} \frac{1}{p_i} (1 - p_i z) \|\xi_i(x) - \xi_i(x^*))\|^2 \\
&\stackrel{(27)}{\leq} z \|\xi(x) - \xi(x^*)\|^2 \\
&\quad + \sum_{i=1}^{n} \frac{1}{n^2} \frac{L_i^2}{p_i} (1 - p_i z) \|x - x^*\|^2 \\
&\stackrel{(28)}{\leq} \left( zL^2 + \sum_{i=1}^{n} \frac{1}{n^2} \frac{L_i^2}{p_i} (1 - p_i z) \right) \|x - x^*\|^2. \quad (29)
\end{aligned}
$$

Since $\xi$ is $\mu$-quasi strongly monotone, then (29) becomes:

$$
\mathbb{E}[\|\xi_v(x) - \xi_v(x^*)\|^2] \leq \left( zL^2 + \sum_{i=1}^{n} \frac{1}{n^2} \frac{L_i^2}{p_i} (1 - p_i z) \right) \frac{1}{\mu} \langle \xi(x), x - x^* \rangle \quad (30)
$$

Thus the expected co-coercivity (EC) is satisfied with

$$
\ell_\xi = \left( zL^2 + \sum_{i=1}^{n} \frac{1}{n^2} \frac{L_i^2}{p_i} (1 - p_i z) \right) \frac{1}{\mu}. \quad (31)
$$

Similar to the proof of Proposition 3.5, using that for $b$-minibatch sampling, $\mathbb{P}[i \in S] = p_i = \frac{b}{n}$ and $z = \frac{n}{b} \frac{b-1}{n-1}$, we obtain:

$$
\ell_\xi = \left( \frac{n}{b} \frac{b-1}{n-1} L^2 + \left( \frac{1}{n} \sum_{i=1}^{n} L_i^2 \right) \frac{1}{b} \frac{n-b}{n-1} \right) \frac{1}{\mu}. \quad (32)
$$

This completes the proof. □

## A.5 Connections of EC to other Assumptions

In this section, we present some propositions not included in the main paper showing properties and connections between classical assumptions and our proposed expected co-coercivity EC.

**Proposition A.8.** In the unconstrained setting, if $\xi$ is $\mu$-quasi strongly monotone, then it is not possible to satisfy the bounded operator assumption that there exists a finite $c$ such that $\mathbb{E}\|\xi_v(x)\|^2 \leq c$ for every $x$ in $\mathbb{R}^d$.

*Proof.* Let us assume that $\mathbb{E}\|\xi_v(x)\|^2 \leq c, \forall x \in \mathbb{R}^d$. Note also that if an operator satisfies the $\mu$-quasi strongly monotone property (2), then by using the Cauchy–Schwarz inequality, it satisfies the error bound condition

$$
\|\xi(x)\| \geq \mu\|x - x^*\|.
$$

By combining the above two inequalities, it holds that:

$$
\mu^2\|x - x^*\|^2 \leq \|\xi(x)\|^2 = \|\mathbb{E}[\xi_v(x)]\|^2 \leq \mathbb{E}\left[\|[\xi_v(x)\|^2\right] \leq c
$$

which means that:

$$
\|x - x^*\|^2 \leq \frac{c}{\mu^2}.
$$

However, for the *unconstrained* stochastic variational inequality problems (1), a point $x$ can be very far from the optimum point $x^*$ and as a result $\|x - x^*\|^2 \geq \frac{c}{\mu^2}$. This leads to a contradiction. □

**Proposition A.9.** In the single-objective optimization when the stochastic problem

$$\min_x \left[ f(x) = \tfrac{1}{n} \sum_{i=1}^n f_i(x) \right]$$

has convex and smooth functions $f_i$, then expected smoothness and expected co-coercivity are equivalent (see last row of Table 1).

*Proof.* For simplicity of exposition, let us focus on single-element sampling.

According to Theorem 2.1.5 in Nesterov [2013], if $f_i$ is convex and $L_i$ smooth, then the following two conditions are equivalent:

$$\|\nabla f_i(x) - \nabla f_i(y)\|^2 \leq 2L_i \left( f_i(x) - f_i(y) - \langle \nabla f_i(y), x - y \rangle \right). \tag{33}$$

$$\|\nabla f_i(x) - \nabla f_i(y)\|^2 \leq L_i \langle \nabla f_i(x) - \nabla f_i(y), x - y \rangle \tag{34}$$

If in the above two condition we select $y = x^*$ and take the expectations with respect to $i$, then we obtain the following two equivalent conditions:

$$\mathbb{E}\left[ \|\nabla f_i(x) - \nabla f_i(x^*)\|^2 \right] \leq 2L_{\max} \left[ f(x) - f(x^*) \right]. \tag{35}$$

$$\mathbb{E}\left[ \|\nabla f_i(x) - \nabla f_i(x^*)\|^2 \right] \leq L_{\max} \langle \nabla f(x), x - x^* \rangle \tag{36}$$

Note that in the above, (35) is the expected smoothness as proposed in Gower et al. [2019] while (36) is our expected co-coercivity (EC) for the single element sampling. Note that for single-objective optimization problems, the operator $\xi$ is simply the gradient vector. As we mentioned in Section 3, in single-objective optimization, a function is $L$–co-coercive if and only if it is convex and $L$-smooth (i.e. $L$-Lipschitz gradients) [Bauschke et al., 2011]. Thus, the co-coercivity constant is equivalent to the smoothness parameter. $\square$

Let us also add a simple remark highlighting the weakness of EC compare to other previously used assumptions in the literature of stochastic algorithms for solving (1).

**Remark A.10.** As we show in Lemma (3.4), by assuming EC we obtain the following bound

$$\mathbb{E}\|\xi_v(x)\|^2 \leq 2\ell_\xi \langle \xi(x), x - x^* \rangle + 2\sigma^2. \tag{37}$$

Let us now compare this bound to the assumption of growth condition $\mathbb{E}\|\xi_i(x)\|^2 \leq c_1\|\xi(x)\|^2 + c_2$ (weakest among the other assumptions).

Note that if an operator $\xi$ is $\ell$–co-coercive, then the growth condition implies:

$$\mathbb{E}\|\xi_i(x)\|^2 \leq c_1 \ell \langle \xi(x), x - x^* \rangle + c_2.$$

This has the same form to the bound (37), obtained by EC. However, the parameters $c_1$ and $c_2$ have unknown values while using EC these parameters are closed-form problem-dependent expressions.

Thus, expected co-coercivity is weaker than the growth condition and at the same time more powerful, as in many case it is not really an assumption, but a condition that is satisfied for free. See for example, Propositions 3.5 and 3.6.

## A.6 Example: Quasi-strongly Monotone Operator that is not Monotone nor Lipschitz

An operator that is $\mu$-quasi strongly monotone may not even be monotone. We now give a simple example of such an operator, with the additional property that it is $L$-co-coercive around $x^*$ but is *not* Lipschitz continuous. This highlights the generality of our convergence results beyond the standard monotone setting. Let $L > \mu > 0$, we define $\xi(x) = x(\frac{L-\mu}{2}\cos(\|x\|_2) + \frac{L+\mu}{2})$. We have that $x^* = 0$ and

$$\langle \xi(x), x - x^* \rangle = \|x - x^*\|_2^2 \left( \frac{L-\mu}{2}\cos(\|x\|_2) + \frac{L+\mu}{2} \right) \geq \mu\|x - x^*\|_2^2, \tag{38}$$

and is thus $\mu$-quasi strongly monotone. However, it is *not* monotone. To see this, let us consider the one dimensional case $x \in \mathbb{R}$. In this case, we get, $\xi(x) = x(\frac{L-\mu}{2}\cos(|x|) + \frac{L+\mu}{2})$. To formally violate the monotonicity inequality, we can for instance consider $x = 2\pi k + \frac{\pi}{2}$ and $x' = 2\pi k$ to get

$$\langle \xi(x) - \xi(x'), x - x' \rangle = (\frac{\pi}{2}\frac{L+\mu}{2} - 2\pi k\frac{L-\mu}{2})\frac{\pi}{2} = \frac{\pi^2}{4}(L + \mu - 4k(L - \mu)). \tag{39}$$

This quantity is negative for $k > \frac{L+\mu}{4(L-\mu)}$.

This operator is also $L$-co-coercive with respect to $x^*$ since,

$$\langle \xi(x), x - x^* \rangle = \|\xi(x)\|_2^2 (\frac{L-\mu}{2}\cos(\|x\|_2) + \frac{L+\mu}{2})^{-1} \geq L^{-1}\|\xi(x)\|_2^2. \tag{40}$$

This operator is *not* Lipschitz continuous for all $x \in \mathbb{R}$, as its derivative is unbounded over $\mathbb{R}$.

## A.7 Example: Co-coercivity for Quadratic Games

For quadratic games, it is relatively easy to characterize co-coercivity.

**Proposition A.11.** If $\xi(x) = Ax$ where $A \in \mathbb{R}^{d \times d}$, we have that $\xi$ is co-coercive if and only if $\langle x, Ax \rangle > 0$, $\forall x \in \mathbb{R}^d \setminus \text{null}(A)$. In that case, we have $\ell = \sup_{\|x\|=1} \frac{\|Ax\|_2^2}{\langle x, Ax \rangle}$.

*Proof.* When $\xi(x) = Ax$, the co-coercivity condition (Definition 3.1)

$$\|A(x - x')\|^2 \leq \ell \langle A(x - x'), x - x' \rangle, \quad \forall x, x' \in \mathbb{R}^d. \tag{41}$$

Now, we can note that the variable of interest is $x - x' \in \mathbb{R}^d$. Thus we get equivalently,

$$\|Ax\|^2 \leq \ell \langle Ax, x \rangle, \quad \forall x \in \mathbb{R}^d. \tag{42}$$

This inequality is valid for any $x \in \mathbb{R}^d$ such that $Ax = 0$. Now, let us consider $x \in \mathbb{R}^d$ such that $Ax \neq 0$.

If there exist $x \in \mathbb{R}^d$ such that $Ax \neq 0$ and $\langle x, Ax \rangle \leq 0$ then, we have that

$$0 < \|Ax\|^2 \leq \ell \langle Ax, x \rangle \leq 0 \tag{43}$$

which is not valid. Thus $\xi$ is *not* co-coercive.

On the other hand, if for all $x \in \mathbb{R}^d$ such that $Ax \neq 0$, we have $\langle x, Ax \rangle > 0$ then, the co-coercivity condition would demand

$$\frac{\|Ax\|^2}{\langle Ax, x \rangle} \leq \ell, \quad \forall x \in \mathbb{R}^d, \, Ax \neq 0. \tag{44}$$

Finally, since the left-hand side of the previous equation is scale invariant ($x \mapsto \lambda x$ does not change the LHS), we have

$$\sup_{x \in \mathbb{R}^d} \frac{\|Ax\|^2}{\langle Ax, x \rangle} = \sup_{x \in \mathbb{R}^d \setminus \{0\}} \frac{\|Ax\|^2}{\langle Ax, x \rangle} = \sup_{\|x\|=1} \frac{\|Ax\|^2}{\langle Ax, x \rangle} \tag{45}$$

So we have $\ell \geq \sup_{\|x\|=1} \frac{\|Ax\|^2}{\langle Ax, x \rangle}$. Because the RHS is continuous in $x$, and the unit ball is a compact, this quantity is achieved. Thus there exists $x \in \mathbb{R}^d$ such that

$$\|Ax\|^2 = \sup_{\|x\|=1} \frac{\|Ax\|^2}{\langle Ax, x \rangle} \langle Ax, x \rangle \tag{46}$$

Thus $\ell \leq \sup_{\|x\|=1} \frac{\|Ax\|^2}{\langle Ax, x \rangle}$, which concludes the proof. $\square$

For instance, a class of quadratic games that are *not* co-coercive are the ones where $A$ is anti-symmetric ($A^\top = -A$). One the other hand, there is a large class of games that are *not* strongly monotone, like for instance any quadratic game induced by a matrix with a non-zero nullspace.

# B    Proofs of Main Convergence Analysis Results

## B.1    Proof of Theorem 4.1

In the main paper, we present the update rule of SGDA in (5). Let us also present here the pseudo-code of SGDA:

---

**Algorithm 2** Stochastic Gradient Descent Ascent (SGDA)

---

**Input:** Starting stepsize $\gamma_0 > 0$. Choose initial points $x^0 \in \mathbb{R}^d$. Distribution $\mathcal{D}$ of samples.
**for** $k = 0, 1, 2, \cdots, K$ **do**
   Sample $v^k \sim \mathcal{D}$
   Set step-size $\gamma_k$ following one of the selected choices (constant, decreasing)
   Set $x^{k+1} = x^k - \gamma_k \xi_{v^k}(x)$
**end for**
**Output:** The last iterate $x^k$

---

Let us present a more general version of Theorem 4.1 that allows convergence with a larger step-size. Due to space limitations, we focus only on the important regime in the main paper, as selecting a larger step-size gives a worse convergence rate.

**Theorem B.1** (Constant Step-size). Assume that $\xi$ is $\mu-$quasi strongly monotone and that $\xi \in EC(\ell_\xi)$. Choose $\alpha_k = \alpha < \frac{1}{\ell_\xi}$ for all k. Then, the iterates of SGDA, given by (5), satisfy:

$$\mathbb{E}\left[\|x^k - x^*\|^2\right] \quad \leq \quad [1 - 2\alpha\mu(1 - \alpha\ell_\xi)]^k \|x^0 - x^*\|^2 + \frac{\alpha\sigma^2}{\mu(1 - \alpha\ell_\xi)} \tag{47}$$

and if $\alpha_k = \alpha \in (0, \frac{1}{2\ell_\xi}]$ then the iterates of SGDA satisfy:

$$\mathbb{E}\left[\|x^k - x^*\|^2\right] \quad \leq \quad (1 - \alpha\mu)^k \|x^0 - x^*\|^2 + \frac{2\alpha\sigma^2}{\mu} \tag{48}$$

*Proof.*

$$\begin{aligned}
\|x^{k+1} - x^*\|^2 &= \left\|x^k - \alpha\xi_{v^k}(x^k) - x^*\right\|^2 \\
&= \|x^k - x^*\|^2 - 2\left\langle x^k - x^*, \alpha\xi_{v^k}(x^k)\right\rangle + \|\alpha\xi_{v^k}(x^k)\|^2 \\
&= \|x^k - x^*\|^2 - 2\alpha\left\langle x^k - x^*, \xi_{v^k}(x^k)\right\rangle + \alpha^2\|\xi_{v^k}(x^k)\|^2 \tag{49}
\end{aligned}$$

By taking expectation condition on $x^k$:

$$\begin{aligned}
\mathbb{E}_\mathcal{D}\left[\|x^{k+1} - x^*\|^2\right] &= \|x^k - x^*\|^2 - 2\alpha\left\langle x^k - x^*, \xi(x^k)\right\rangle + \alpha^2\mathbb{E}_\mathcal{D}\left[\|\xi_{v^k}(x^k)\|^2\right] \\
&\overset{Lemma\ 3.4}{\leq} \|x^k - x^*\|^2 - 2\alpha\left\langle x^k - x^*, \xi(x^k)\right\rangle + 2\alpha^2\ell_\xi\left\langle x^k - x^*, \xi(x^k)\right\rangle + 2\alpha^2\sigma^2 \\
&\overset{(2),\alpha<\frac{1}{\ell_\xi}}{\leq} \|x^k - x^*\|^2 - 2\alpha\mu(1 - \alpha\ell_\xi)\|x^k - x^*\|^2 + 2\alpha^2\sigma^2 \tag{50}
\end{aligned}$$

Recursively applying the above and summing up the resulting geometric series gives:

$$\begin{aligned}
\mathbb{E}\left[\|x^k - x^*\|^2\right] &\leq [1 - 2\alpha\mu(1 - \alpha\ell_\xi)]^k \|x^0 - x^*\|^2 + 2\sum_{j=0}^{k-1}(1 - 2\alpha\mu(1 - \alpha\ell_\xi))^j \alpha^2\sigma^2 \\
&\leq [1 - 2\alpha\mu(1 - \alpha\ell_\xi)]^k \|x^0 - x^*\|^2 + \frac{\alpha\sigma^2}{\mu(1 - \alpha\ell_\xi)} \tag{51}
\end{aligned}$$

If we further take $\alpha \leq \frac{1}{2\ell_\xi}$ then (50) becomes:

$$\mathbb{E}_\mathcal{D}\left[\|x^{k+1} - x^*\|^2\right] \quad \leq \quad (1 - \alpha\mu)\|x^k - x^*\|^2 + 2\alpha^2\sigma^2 \tag{52}$$

and by recursively applying the above and summing up the resulting geometric series gives:

$$\mathbb{E}\left[\|x^k - x^*\|^2\right] \quad \leq \quad (1 - \alpha\mu)^k\|x^0 - x^*\|^2 + \frac{2\alpha\sigma^2}{\mu} \tag{53}$$

$\square$

**Comment on the convergence deterministic Gradient Descent Ascent:** In the main paper, to highlight the generality of Theorem 4.1, we present Corollary 4.2 on the convergence of deterministic gradient descent ascent. Let us provide some more details of how one can obtain such a result through Proposition 3.5.

Let us select the sampling vector $v = (1, 1, \ldots, 1) \in R^n$ with probability 1 in each step. Note that this is still a sampling vector as $\mathbb{E}_{\mathcal{D}}[v_i] = 1$. In this case, at iteration $k$, $\xi_{v^k}(x^k) := \frac{1}{n}\sum_{i=1}^n v_i\xi_i(x^k) \stackrel{v_i=1}{=} \frac{1}{n}\sum_{i=1}^n \xi_i(x^k) = \xi(x^k)$ and the update rule becomes equivalent to the deterministic GDA:

$$x^{k+1} = x^k - \alpha_k\xi(x^k).$$

In addition, by Proposition 3.5 we have that if $|S| = n$ with probability one (each iteration of SGDA uses a full batch gradient), then $\ell_\xi = \ell$ and $\sigma^2 = 0$. Thus, by combining (7) of Theorem 4.1 with Proposition 3.5 we obtain the convergence given in Corollary 4.2 for the deterministic gradient descent ascent. We highlight that for this case, the expected co-coercivity condition (EC) is equivalent to assuming that operator $\xi$ is $\ell$-co-coercive.

## B.2 Proof of Theorem 4.3

**Theorem B.2.** Assume $\xi$ is $\mu$-quasi-strongly monotone and that $\xi \in EC(\ell_\xi)$. Let $\mathcal{K} := \ell_\xi/\mu$ and let

$$\alpha_k = \begin{cases} \dfrac{1}{2\ell_\xi} & \text{for} \quad k \leq 4\lceil\mathcal{K}\rceil \\ \dfrac{2k+1}{(k+1)^2\mu} & \text{for} \quad k > 4\lceil\mathcal{K}\rceil. \end{cases} \tag{54}$$

If $k \geq 4\lceil\mathcal{K}\rceil$, then iterates of SGDA, given by (5) satisfy:

$$\mathbb{E}\|x^k - x^*\|^2 \leq \frac{\sigma^2}{\mu^2}\frac{8}{k} + \frac{16\lceil\mathcal{K}\rceil^2}{e^2 k^2}\|x^0 - x^*\|^2 = O\left(\frac{1}{k}\right). \tag{55}$$

*Proof.* Let $\alpha_k := \frac{2k+1}{(k+1)^2\mu}$ and let $k^*$ be an integer that satisfies $\alpha_{k^*} \leq \frac{1}{2\ell_\xi}$. Note that $\alpha_k$ is decreasing in $k$ and consequently $\alpha_k \leq \frac{1}{2\ell_\xi}$ for all $k \geq k^*$. This in turn guarantees that (52) holds for all $k \geq k^*$ with $\alpha_k$ in place of $\alpha$, that is

$$\mathbb{E}_{\mathcal{D}}\left[\|x^{k+1} - x^*\|^2\right] \quad \leq \quad (1 - \alpha_k\mu)\|x^k - x^*\|^2 + 2\alpha_k^2\sigma^2 \tag{56}$$

Hence, if we take expectations and replace $\alpha_k := \frac{2k+1}{(k+1)^2\mu}$ then

$$\mathbb{E}\|x^{k+1} - x^*\|^2 \leq \frac{k^2}{(k+1)^2}\mathbb{E}\|x^k - x^*\|^2 + \frac{2\sigma^2}{\mu^2}\frac{(2k+1)^2}{(k+1)^4}. \tag{57}$$

Multiplying both sides by $(k+1)^2$ we obtain

$$(k+1)^2\mathbb{E}\|x^{k+1} - x^*\|^2 \quad \leq \quad k^2\mathbb{E}\|x^k - x^*\|^2 + \frac{2\sigma^2}{\mu^2}\left(\frac{2k+1}{k+1}\right)^2$$

$$\leq \quad k^2\mathbb{E}\|x^k - x^*\|^2 + \frac{8\sigma^2}{\mu^2},$$

where the second inequality holds because $\frac{2k+1}{k+1} < 2$. Rearranging and summing from $t = k^* \ldots k$ we obtain:

$$\sum_{t=k^*}^k \left[(t+1)^2\mathbb{E}\|x^{k+1} - x^*\|^2 - t^2\mathbb{E}\|x^k - x^*\|^2\right] \leq \sum_{t=k^*}^k \frac{8\sigma^2}{\mu^2}. \tag{58}$$

Using telescopic cancellation gives

$$(k+1)^2 \mathbb{E}\|x^{k+1} - x^*\|^2 \le (k^*)^2 \mathbb{E}\|x^{k^*} - x^*\|^2 + \frac{8\sigma^2(k-k^*)}{\mu^2}.$$

Dividing the above by $(k+1)^2$ gives

$$\mathbb{E}\|x^{k+1} - x^*\|^2 \le \frac{(k^*)^2}{(k+1)^2}\mathbb{E}\|x^{k^*} - x^*\|^2 + \frac{8\sigma^2(k-k^*)}{\mu^2(k+1)^2}. \tag{59}$$

For $k \le k^*$ we have that (53) holds with $\alpha_k = \frac{1}{2\ell_\xi}$, which combined with (59), gives

$$
\begin{aligned}
\mathbb{E}\|x^{k+1} - x^*\|^2 \quad \le \quad & \frac{(k^*)^2}{(k+1)^2}\left(1 - \frac{\mu}{2\ell_\xi}\right)^{k^*}\|x^0 - x^*\|^2 \\
+ \quad & \frac{\sigma^2}{\mu^2(k+1)^2}\left(8(k-k^*) + \frac{(k^*)^2}{\mathcal{K}}\right).
\end{aligned}
\tag{60}
$$

Choosing $k^*$ that minimizes the second line of the above gives $k^* = 4\lceil\mathcal{K}\rceil$, which when inserted into (60) becomes

$$
\begin{aligned}
\mathbb{E}\|x^{k+1} - x^*\|^2 \quad \le \quad & \frac{16\lceil\mathcal{K}\rceil^2}{(k+1)^2}\left(1 - \frac{1}{2\mathcal{K}}\right)^{4\lceil\mathcal{K}\rceil}\|x^0 - x^*\|^2 \\
& + \frac{\sigma^2}{\mu^2}\frac{8(k - 2\lceil\mathcal{K}\rceil)}{(k+1)^2} \\
\le \quad & \frac{16\lceil\mathcal{K}\rceil^2}{e^2(k+1)^2}\|x^0 - x^*\|^2 + \frac{\sigma^2}{\mu^2}\frac{8}{k+1},
\end{aligned}
\tag{61}
$$

where we have used that $\left(1 - \frac{1}{2x}\right)^{4x} \le e^{-2}$ for all $x \ge 1$. $\qquad\square$

## B.3  Proof of Theorem 5.1

Before providing the proof of Theorem 5.1, let us present the definitions of quasi-strong convexity and expected smoothness condition, together with a lemma that provides a bound to the expected norm of the stochastic gradients when a function satisfies the expected smoothness. Recall that in the main paper, we assume that the Hamiltonian function $\mathcal{H}(x)$ is quasi-strongly convex and $\mathcal{L}$—expected smooth (satisfies the expected smoothness condition). Thus, these assumptions are vital for the convergence guarantees of SCO presented in Section 5.

**Technical Background on Optimization.**    Let us consider the optimization problem

$$x^* = \operatorname{argmin}_{x \in \mathbb{R}^d}\left[f(x) = \frac{1}{n}\sum_{i=1}^{n} f_i(x)\right], \tag{62}$$

where each $f_i : \mathbb{R}^d \to \mathbb{R}$ is smooth and $f$ has a unique global minimizer $x^*$.

**Definition B.3** (Quasi-strong convexity). We say that a function $f : \mathbb{R}^d \to \mathbb{R}$ is $\mu$–strongly quasi-convex [Karimi et al., 2016, Necoara et al., 2018] if there is $\mu > 0$ such that:

$$f(x^*) \ge f(x) + \langle \nabla f(x), x^* - x\rangle + \frac{\mu}{2}\|x^* - x\|^2 \tag{63}$$

for all $x \in \mathbb{R}^d$. Here $x^*$ is the global minimizer of $f$[a].

Note that we have already presented the expected smoothness condition in Table 1. Below we present its formal definition. For this definition we use the stochastic reformulation of the finite-sum problem $f(x) = \frac{1}{n}\sum_{i=1}^{n} f_i(x)$. That is, we define: $f_v(x) := \frac{1}{n}\sum_{i=1}^{n} v_i f_i(x)$ where $v \sim \mathcal{D}$ is a random sampling vector (see Section 2 for more details on sampling vectors).

---

[a]In our setting we assume that $x^*$ is unique, but in the more general setting, $x^*$ is the projection of point $x$ onto the solution set $X^*$ minimizing $f$.

**Definition B.4** (Expected Smoothness). We say that $f$ is $\mathcal{L}$—expected smooth with respect to a distribution $\mathcal{D}$ if there exists $\mathcal{L} = \mathcal{L}(f, \mathcal{D}) > 0$ such that

$$\mathbb{E}_{\mathcal{D}} \left[ \|\nabla f_v(x) - \nabla f_v(x^*)\|^2 \right] \leq 2\mathcal{L}(f(x) - f(x^*)), \tag{64}$$

for all $x \in \mathbb{R}^d$.

In the next lemma, by assuming that a function $f(x) = \frac{1}{n} \sum_{i=1}^{n} f_i(x)$ satisfies the expected smoothness we are able to bound the expected norm of its stochastic gradients. This is precisely the result we use in our proofs on the convergence of SCO, to upper bound $\mathbb{E}_{\mathcal{D}} \left[ \|\nabla \mathcal{H}_{v^k, u^k}(x^k)\|^2 \right]$. This bound allows us to avoid the much stronger bounded gradient or bounded variance assumptions.

**Lemma B.5** (Lemma 2.4 in Gower et al. [2019]). If $f$ is $\mathcal{L}$—expected smooth, then

$$\mathbb{E}_{\mathcal{D}} \left[ \|\nabla f_v(x)\|^2 \right] \leq 4\mathcal{L}(f(x) - f(x^*)) + 2\sigma^2. \tag{65}$$

where $\sigma^2 := \mathbb{E}_{\mathcal{D}}[\|\nabla f_v(x^*)\|^2]$.

*Proof.* The proof can be easily obtained by following the same steps of the proof of Lemma 3.4. See also the proof of Lemma 2.4 in Gower et al. [2019]. $\square$

Let us now present a more general version of Theorem 5.1 that allows convergence with a larger step-size $\alpha$. Due to space limitations, in the main paper, we focus only on the important regime of step-size $\alpha$ as selecting a larger step-size gives a worse convergence rate.

**Theorem B.6** (Constant Step-size). Assume $\xi$ is $\mu$-quasi-strongly monotone with $\mu \geq 0$ and that $\xi \in EC(\ell_\xi)$. Let us also assume that the Hamiltonian function $\mathcal{H}$ is $\mu_{\mathcal{H}}$-quasi strongly convex and $\mathcal{L}_{\mathcal{H}}$-expected smooth. Then, for $\gamma_k = \gamma \leq 1/4\mathcal{L}_{\mathcal{H}}$ and $\alpha_k = \alpha < 1/2\ell_\xi$ it holds that :

$$\mathbb{E} \left[ \|x^k - x^*\|^2 \right] \leq (1 - \gamma\mu_{\mathcal{H}} - 2\alpha\mu + 4\alpha^2\ell_\xi\mu)^k \|x^0 - x^*\|^2 + \frac{4[\alpha^2\sigma^2 + \gamma^2\sigma_{\mathcal{H}}^2]}{\gamma\mu_{\mathcal{H}} + 2\alpha\mu - 4\alpha^2\ell_\xi\mu}, \tag{66}$$

and if $\alpha_k = \alpha \in (0, \frac{1}{4\ell_\xi}]$ then the iterates of SCO satisfy:

$$\mathbb{E} \left[ \|x^k - x^*\|^2 \right] \leq (1 - \gamma\mu_{\mathcal{H}} - \alpha\mu)^k \|x^0 - x^*\|^2 + \frac{4[\alpha^2\sigma^2 + \gamma^2\sigma_{\mathcal{H}}^2]}{\gamma\mu_{\mathcal{H}} + \alpha\mu}. \tag{67}$$

If $\mu = 0$, that is $\xi$ only satisfies the variational stability condition $\langle \xi(x), x - x^* \rangle \geq 0$, then

$$\mathbb{E} \left[ \|x^k - x^*\|^2 \right] \leq (1 - \gamma\mu_{\mathcal{H}})^k \|x^0 - x^*\|^2 + \frac{4[\alpha^2\sigma^2 + \gamma^2\sigma_{\mathcal{H}}^2]}{\gamma\mu_{\mathcal{H}}}. \tag{68}$$

*Proof.*

$$
\begin{aligned}
\|x^{k+1} - x^*\|^2 &= \left\| x^k - \alpha\xi_{v^k}(x^k) - \gamma\nabla\mathcal{H}_{v^k, u^k}(x^k) - x^* \right\|^2 \\
&= \|x^k - x^*\|^2 - 2\left\langle x^k - x^*, \alpha\xi_{v^k}(x^k) + \gamma\nabla\mathcal{H}_{v^k, u^k}(x^k) \right\rangle \\
&\quad + \|\alpha\xi_{v^k}(x^k) + \gamma\nabla\mathcal{H}_{v^k, u^k}(x^k)\|^2 \\
&= \|x^k - x^*\|^2 - 2\alpha\left\langle x^k - x^*, \xi_{v^k}(x^k) \right\rangle - 2\gamma\left\langle x^k - x^*, \nabla\mathcal{H}_{v^k, u^k}(x^k) \right\rangle \\
&\quad + \|\alpha\xi_{v^k}(x^k) + \gamma\nabla\mathcal{H}_{v^k, u^k}(x^k)\|^2 \\
&\overset{\text{Young's}}{\leq} \|x^k - x^*\|^2 - 2\alpha\left\langle x^k - x^*, \xi_{v^k}(x^k) \right\rangle - 2\gamma\left\langle x^k - x^*, \nabla\mathcal{H}_{v^k, u^k}(x^k) \right\rangle \\
&\quad + 2\alpha^2 \left\| \xi_{v^k}(x^k) \right\|^2 + 2\gamma^2 \|\nabla\mathcal{H}_{v^k, u^k}(x^k)\|^2 \tag{69}
\end{aligned}
$$

By taking expectation condition on $x^k$:

$$\mathbb{E}_{\mathcal{D}}\left[\|x^{k+1} - x^*\|^2\right] \leq \|x^k - x^*\|^2 - 2\alpha \left\langle x^k - x^*, \xi(x^k) \right\rangle - 2\gamma \left\langle x^k - x^*, \nabla\mathcal{H}(x^k) \right\rangle$$
$$+ 2\alpha^2 \mathbb{E}_{\mathcal{D}}\left[\left\|\xi_{v^k}(x^k)\right\|^2\right] + 2\gamma^2 \mathbb{E}_{\mathcal{D}}\left[\|\nabla\mathcal{H}_{v^k,u^k}(x^k)\|^2\right]$$

$$\overset{(63)}{\leq} \|x^k - x^*\|^2 - 2\alpha \left\langle x^k - x^*, \xi(x^k) \right\rangle - 2\gamma[\mathcal{H}(x^k) - \mathcal{H}(x^*)] - \gamma\mu_{\mathcal{H}}\|x^k - x^*\|^2$$
$$+ 2\alpha^2 \mathbb{E}_{\mathcal{D}}\left[\left\|\xi_{v^k}(x^k)\right\|^2\right] + 2\gamma^2 \mathbb{E}_{\mathcal{D}}\left[\|\nabla\mathcal{H}_{v^k,u^k}(x^k)\|^2\right]$$

$$\overset{(65)}{\leq} \|x^k - x^*\|^2 - 2\alpha \left\langle x^k - x^*, \xi(x^k) \right\rangle - 2\gamma[\mathcal{H}(x^k) - \mathcal{H}(x^*)] - \gamma\mu_{\mathcal{H}}\|x^k - x^*\|^2$$
$$+ 2\alpha^2 \mathbb{E}_{\mathcal{D}}\left[\left\|\xi_{v^k}(x^k)\right\|^2\right] + 8\gamma^2 \mathcal{L}_{\mathcal{H}}(\mathcal{H}(x) - \mathcal{H}(x^*)) + 4\gamma^2\sigma_{\mathcal{H}}^2$$

$$\overset{(16)}{\leq} \|x^k - x^*\|^2 - 2\alpha \left\langle x^k - x^*, \xi(x^k) \right\rangle - 2\gamma[\mathcal{H}(x^k) - \mathcal{H}(x^*)] - \gamma\mu_{\mathcal{H}}\|x^k - x^*\|^2$$
$$+ 4\alpha^2\ell_\xi \langle \xi(x), x - x^* \rangle + 4\alpha^2\sigma^2 + 8\gamma^2 \mathcal{L}_{\mathcal{H}}(\mathcal{H}(x) - \mathcal{H}(x^*)) + 4\gamma^2\sigma_{\mathcal{H}}^2 \quad (70)$$

Recall that $\xi$ is $\mu$-quasi strongly monotone, $\left\langle x^k - x^*, \xi(x^k) \right\rangle \geq \mu\|x^k - x^*\|^2$. Thus, for $\alpha_k < \frac{1}{2\ell_\xi}$, it holds that:

$$(-2\alpha_k + 4\alpha_k^2\ell_\xi)\langle \xi(x^k), x - x^* \rangle \leq (-2\alpha_k + 4\alpha_k^2\ell_\xi)\mu\|x^k - x^*\|^2,$$

and the inequality (70) takes the following form:

$$\mathbb{E}_{\mathcal{D}}\left[\|x^{k+1} - x^*\|^2\right] \leq (1 - \gamma\mu_{\mathcal{H}})\|x^k - x^*\|^2 + (-2\alpha + 4\alpha^2\ell_\xi)\mu\|x^k - x^*\|^2$$
$$+ (-2\gamma + 8\gamma_k^2\mathcal{L}_{\mathcal{H}})[\mathcal{H}(x^k) - \mathcal{H}(x^*)] + 4\alpha^2\sigma^2 + 4\gamma^2\sigma_{\mathcal{H}}^2$$
$$= (1 - \gamma\mu_{\mathcal{H}} - 2\alpha\mu + 4\alpha^2\ell_\xi\mu)\|x^k - x^*\|^2$$
$$+ (-2\gamma + 8\gamma^2\mathcal{L}_{\mathcal{H}})[\mathcal{H}(x^k) - \mathcal{H}(x^*)] + 4\alpha^2\sigma^2 + 4\gamma^2\sigma_{\mathcal{H}}^2$$
$$\overset{\gamma_k < \frac{1}{4\mathcal{L}_{\mathcal{H}}}}{\leq} (1 - \gamma\mu_{\mathcal{H}} - 2\alpha\mu + 4\alpha^2\ell_\xi\mu)\|x^k - x^*\|^2 + 4[\alpha^2\sigma^2 + \gamma^2\sigma_{\mathcal{H}}^2]$$
$$(71)$$

By taking expectations again and by recursively applying the above and summing up the resulting geometric series gives:

$$\mathbb{E}\left[\|x^k - x^*\|^2\right] \leq (1 - \gamma\mu_{\mathcal{H}} - 2\alpha\mu + 4\alpha^2\ell_\xi\mu)^k\|x^0 - x^*\|^2 + \frac{4[\alpha^2\sigma^2 + \gamma^2\sigma_{\mathcal{H}}^2]}{\gamma\mu_{\mathcal{H}} + 2\alpha\mu - 4\alpha^2\ell_\xi\mu}$$

If we further assume that $\alpha \leq \frac{1}{4\ell_\xi}$ then $1 - \gamma\mu_{\mathcal{H}} - 2\alpha\mu_g + 4\alpha^2\ell_\xi\mu_g \leq 1 - \gamma\mu_{\mathcal{H}} - \alpha\mu$ and the iterates of SCO satisfy:

$$\mathbb{E}\left[\|x^k - x^*\|^2\right] \leq (1 - \gamma\mu_{\mathcal{H}} - \alpha\mu)^k\|x^0 - x^*\|^2 + \frac{4[\alpha^2\sigma^2 + \gamma^2\sigma_{\mathcal{H}}^2]}{\gamma\mu_{\mathcal{H}} + \alpha\mu}. \quad (72)$$

In addition, if $\mu = 0$, that is $\xi$ only satisfies the variational stability condition $\langle \xi(x), x - x^* \rangle \geq 0$, then

$$\mathbb{E}\left[\|x^k - x^*\|^2\right] \leq (1 - \gamma\mu_{\mathcal{H}})^k\|x^0 - x^*\|^2 + \frac{4[\alpha^2\sigma^2 + \gamma^2\sigma_{\mathcal{H}}^2]}{\gamma\mu_{\mathcal{H}}}. \quad (73)$$

This completes the proof. $\qquad\square$

**On Deterministic Consensus Optimization.** In Corollary 5.2 we show the convergence of Deterministic CO as special case of our main Theorem. Here we provide few more details to understand exactly this convergence.

Let us select the sampling vectors $v = u = (1, 1, \ldots, 1) \in R^n$ with probability 1 in each step. Note that these are still sampling vectors as $\mathbb{E}_{\mathcal{D}}[v_i] = \mathbb{E}_{\mathcal{D}}[u_i] = 1$. In this case, at iteration $k$, $\xi_{v^k}(x^k) := \frac{1}{n}\sum_{i=1}^n v_i\xi_i(x^k) \overset{v_i=1}{=} \frac{1}{n}\sum_{i=1}^n \xi_i(x^k) = \xi(x^k)$ and $\nabla\mathcal{H}_{u,v}(x) =$

$\frac{1}{2} \left[ \mathbf{J}_u^\top(x)\xi_v(x) + \mathbf{J}_v^\top(x)\xi_u(x) \right] \overset{v_i=1,u_i=1}{=} \frac{1}{2} \left[ \mathbf{J}^\top(x)\xi(x) + \mathbf{J}^\top(x)\xi(x) \right] = \mathbf{J}^\top(x)\xi(x) = \nabla\mathcal{H}(x).$

Thus, the update rule becomes equivalent to the deterministic CO:

$$x^{k+1} = x^k - \alpha\xi(x^k) - \gamma\nabla\mathcal{H}(x^k).$$

In addition, by Proposition 3.5 we have that if $|S| = n$ with probability one, then $\ell_\xi = \ell$ and $\sigma^2 = 0$. Using also the properties of expected smoothness (see Proposition 3.8 in Gower et al. [2019]) we have that $\mathcal{L}_\mathcal{H} = L_\mathcal{H}$, where $L_\mathcal{H}$ is the smoothness parameter of the Hamiltonian function, and $\sigma_\mathcal{H}^2 = 0$. By simply substituting these values to the main theorem we are able to obtain the convergence result presented in Corollary 5.2.

**Convergence of SGDA and SHGD as special cases of our Analysis.** As we mentioned in the main paper, SCO is a weighted combination of SGDA (Algorithm 2) and the stochastic Hamiltonian gradient descent (SHGD) of Loizou et al. [2020]. Thus, it is clear, that if one selects $\alpha^k = 0, \forall k > 0$ then the method is equivalent to SHGD and if $\gamma^k = 0, \forall k > 0$ then the method becomes equivalent to the SGDA. Here we highlight that in these cases the Young's inequality used in (69) of the proof of main theorem is not necessary. This is exactly why by specifying the update rule to SHGD we are able to have convergence using the larger bound on the step-size $\gamma \leq 1/2\mathcal{L}_\mathcal{H}$ (see Corollary 5.3). Following similar argument the convergence of SGDA presented in Theorem 4.1 can also obtained as special case of Theorem 5.1.

## B.4   Proof of Theorem 5.4

**Theorem B.7.** Assume $\xi$ is $\mu$-quasi-strongly monotone and that $\xi \in EC(\ell_\xi)$. Assume that the Hamiltonian function $\mathcal{H}$ is $\mu_\mathcal{H}$-quasi strongly convex and $\mathcal{L}_\mathcal{H}$-expected smooth. Let $\alpha_k = \gamma_k$, $\psi = \max\{\ell_\xi, \mathcal{L}_\mathcal{H}\}$ and $k^* := 8\frac{\psi}{\mu_\mathcal{H}+\mu}$. Let also,

$$\gamma_k = \begin{cases} \dfrac{1}{4\psi} & \text{for} \quad k \leq \lceil k^* \rceil \\[3mm] \dfrac{2k+1}{(k+1)^2[\mu_\mathcal{H}+\mu]} & \text{for} \quad k > \lceil k^* \rceil. \end{cases} \tag{74}$$

If $k \geq \lceil k^* \rceil$, then SCO iterates satisfy:

$$\mathbb{E}\|x^k - x^*\|^2 \leq \frac{\sigma_\mathcal{H}^2 + \sigma^2}{[\mu + \mu_\mathcal{H}]^2}\frac{16}{k} + \frac{(k^*)^2}{e^2 k^2}\|x^0 - x^*\|^2 = O\left(\frac{1}{k}\right) \tag{75}$$

If $\mu = 0$, that is $\xi$ only satisfies the variational stability condition $\langle \xi(x), x - x^* \rangle \geq 0$, then SCO is still able to converge sublinearly with $O\left(\frac{1}{k}\right)$, to $x^*$.

*Proof.* Let $\gamma_k := \frac{2k+1}{(k+1)^2[\mu_\mathcal{H}+\mu]}$ and let $k^*$ be an integer that satisfies

$$\gamma_{k^*} \leq \min\left\{\frac{1}{4\ell_\xi}, \frac{1}{4\mathcal{L}_\mathcal{H}}\right\} = \frac{1}{4\psi}.$$

Note that $\gamma_k$ is decreasing in $k$ and consequently $\gamma_k \leq \frac{1}{4\psi}$ for all $k \geq k^*$. This in turn guarantees that (71) holds for all $k \geq k^*$ with $\gamma_k$ in place of $\gamma$ and $\alpha_k = \gamma_k$ in place of $\alpha$, that is

$$\mathbb{E}_\mathcal{D}\left[\|x^{k+1} - x^*\|^2\right] \leq (1 - \gamma_k\mu_\mathcal{H} - 2\gamma_k\mu + 4\gamma_k^2\ell_\xi\mu)\|x^k - x^*\|^2 + 4[\gamma_k^2\sigma^2 + \gamma_k^2\sigma_\mathcal{H}^2]$$

and since $\gamma_k \leq \frac{1}{4\ell_\xi}$ we obtain:

$$\mathbb{E}_\mathcal{D}\left[\|x^{k+1} - x^*\|^2\right] \leq (1 - \gamma_k[\mu_\mathcal{H} + \mu])\|x^k - x^*\|^2 + 4\gamma_k^2[\sigma^2 + \sigma_\mathcal{H}^2]. \tag{76}$$

For simplicity of presentation let us denote $\bar{\mu} = [\mu_\mathcal{H} + \mu]$ and $\bar{\sigma}^2 = [\sigma^2 + \sigma_\mathcal{H}^2]$ then (76) can be written as:

$$\mathbb{E}_\mathcal{D}\left[\|x^{k+1} - x^*\|^2\right] \leq (1 - \gamma_k\bar{\mu})\|x^k - x^*\|^2 + 4\gamma_k^2\bar{\sigma}^2. \tag{77}$$

Now let us follow similar steps to the proof of Theorem 4.3.

By taking expectations and replacing $\gamma_k := \frac{2k+1}{(k+1)^2[\mu_{\mathcal{H}}+\mu]} = \frac{2k+1}{(k+1)^2\bar{\mu}}$ we obtain,

$$\mathbb{E}\|x^{k+1} - x^*\|^2 \leq \frac{k^2}{(k+1)^2}\mathbb{E}\|x^k - x^*\|^2 + \frac{4\bar{\sigma}^2}{\bar{\mu}^2}\frac{(2k+1)^2}{(k+1)^4}. \tag{78}$$

Multiplying both sides by $(k+1)^2$ we obtain

$$\begin{aligned} (k+1)^2\mathbb{E}\|x^{k+1} - x^*\|^2 &\leq k^2\mathbb{E}\|x^k - x^*\|^2 + \frac{4\bar{\sigma}^2}{\bar{\mu}^2}\left(\frac{2k+1}{k+1}\right)^2 \\ &\leq k^2\mathbb{E}\|x^k - x^*\|^2 + \frac{16\bar{\sigma}^2}{\bar{\mu}^2}, \end{aligned}$$

where the second inequality holds because $\frac{2k+1}{k+1} < 2$. Rearranging and summing from $t = k^* \ldots k$ we obtain:

$$\sum_{t=k^*}^{k}\left[(t+1)^2\mathbb{E}\|x^{k+1} - x^*\|^2 - t^2\mathbb{E}\|x^k - x^*\|^2\right] \leq \sum_{t=k^*}^{k}\frac{16\bar{\sigma}^2}{\bar{\mu}^2}. \tag{79}$$

Using telescopic cancellation gives

$$(k+1)^2\mathbb{E}\|x^{k+1} - x^*\|^2 \leq (k^*)^2\mathbb{E}\|x^{k^*} - x^*\|^2 + \frac{8\bar{\sigma}^2(k-k^*)}{\bar{\mu}^2}.$$

Dividing the above by $(k+1)^2$ gives

$$\mathbb{E}\|x^{k+1} - x^*\|^2 \leq \frac{(k^*)^2}{(k+1)^2}\mathbb{E}\|x^{k^*} - x^*\|^2 + \frac{8\bar{\sigma}^2(k-k^*)}{\bar{\mu}^2(k+1)^2}. \tag{80}$$

At this point note that for $k \leq k^*$ we have that (77) holds and by using our step-size selection $\gamma_k = \gamma = \frac{1}{4\psi}$ for $k \leq \lceil k^*\rceil$ we obtain

$$\mathbb{E}_{\mathcal{D}}\left[\|x^{k+1} - x^*\|^2\right] \leq (1 - \gamma\bar{\mu})\|x^k - x^*\|^2 + 4\gamma^2\bar{\sigma}^2, \tag{81}$$

which by taking expectations again and by recursively applying the above and summing up the resulting geometric series gives (for $k \leq k^*$):

$$\mathbb{E}\left[\|x^{k+1} - x^*\|^2\right] \leq (1 - \gamma\bar{\mu})^k\|x^0 - x^*\|^2 + \frac{4\gamma\bar{\sigma}^2}{\bar{\mu}}, \tag{82}$$

Thus, for $k \leq k^*$ we have that (82) holds with $\gamma_k = \frac{1}{4\psi}$, which combined with (80), gives

$$\begin{aligned} \mathbb{E}\|x^{k+1} - x^*\|^2 &\leq \frac{(k^*)^2}{(k+1)^2}\left(1 - \frac{\bar{\mu}}{4\psi}\right)^{k^*}\|x^0 - x^*\|^2 \\ &+ \frac{\bar{\sigma}^2}{\bar{\mu}^2(k+1)^2}\left(16(k-k^*) + \frac{(k^*)^2\bar{\mu}}{\psi}\right). \end{aligned} \tag{83}$$

Choosing $k^*$ that minimizes the second line of the above gives $k^* = 8\frac{\psi}{\bar{\mu}}$, which when inserted into (83) becomes

$$\begin{aligned} \mathbb{E}\|x^{k+1} - x^*\|^2 &\leq \frac{(k^*)^2}{(k+1)^2}\left(1 - \frac{2}{k^*}\right)^{k^*}\|x^0 - x^*\|^2 \\ &+ \frac{\bar{\sigma}^2}{\bar{\mu}^2(k+1)^2}8\,(2k-k^*) \\ &\leq \frac{(k^*)^2}{(k+1)^2e^2}\|x^0 - x^*\|^2 + \frac{\bar{\sigma}^2}{\bar{\mu}^2(k+1)^2}8\,(2k-k^*) \\ &\leq \frac{(k^*)^2}{(k+1)^2e^2}\|x^0 - x^*\|^2 + \frac{\bar{\sigma}^2}{\bar{\mu}^2}\frac{16}{k+1}. \end{aligned} \tag{84}$$

where in the second inequality we have used that $\left(1 - \frac{1}{2x}\right)^{4x} \leq \frac{1}{e^2}$ for all $x \geq 1$ and in the last inequality we used that $\frac{2k-k^*}{k+1} \leq \frac{2k}{k+1} \leq 2$.

Thus by replacing $\bar{\mu} = [\mu_{\mathcal{H}} + \mu]$ and $\bar{\sigma}^2 = [\sigma^2 + \sigma_{\mathcal{H}}^2]$ we obtain:

$$\mathbb{E}\|x^k - x^*\|^2 \leq \frac{\sigma_{\mathcal{H}}^2 + \sigma^2}{[\mu + \mu_{\mathcal{H}}]^2} \frac{16}{k} + \frac{(k^*)^2}{e^2 k^2} \|x^0 - x^*\|^2 = O\left(\frac{1}{k}\right).$$

As we mentioned in the statement of the Theorem, if $\mu = 0$, that is $\xi$ only satisfies the variational stability condition $\langle \xi(x), x - x^* \rangle \geq 0$, then SCO is still able to converge sublinearly with $O\left(\frac{1}{k}\right)$, to $x^*$. In this case, the proof will be exactly the same as above but $k^* := 8\frac{\psi}{\mu_{\mathcal{H}}}$ and $\gamma_k = \frac{2k+1}{(k+1)^2 \mu_{\mathcal{H}}}$ for $k > \lceil k^* \rceil$. And the convergence will be $\mathbb{E}\|x^k - x^*\|^2 \leq \frac{\sigma_{\mathcal{H}}^2 + \sigma^2}{\mu_{\mathcal{H}}^2} \frac{16}{k} + \frac{(k^*)^2}{e^2 k^2} \|x^0 - x^*\|^2 = O\left(\frac{1}{k}\right)$. $\qquad \square$

## C On Experiments

### C.1 Properties of Hamiltonian Function for Quadratic Games

In the next proposition, we explain how the assumptions on the Hamiltonian function used in the main theorems of Section 5 (convergence analysis of SCO) are satisfied for the quadratic min-max problems.

> **Proposition C.1.** For quadratic games of the form (13) with $\mathbf{A}_i$ and $\mathbf{C}_i$ symmetric with at least one solution $x^*$, the Hamiltonian function $\mathcal{H}(x)$ is a $L_\mathcal{H}$-smooth and $\mu_\mathcal{H}$–quasi-strongly convex quadratic function with constants $L_\mathcal{H} = \sigma_{\max}^2(\mathbf{J})$ and $\mu_\mathcal{H} = \sigma_{\min}^2(\mathbf{J})$ where $\sigma_{\max}$ and $\sigma_{\min}$ are the maximum and minimum non-zero singular values of $\mathbf{J}$, and $\mathbf{J} = \nabla \xi$ is the Jacobian matrix of the game.

*Proof.* Our approach follows closely the proof of Proposition 4.3 of Loizou et al. [2020] where the properties of the Hamiltonian function for stochastic bilinear games were presented. Let $\mathbf{A} = \frac{1}{n} \sum_{i=1}^n \mathbf{A}_i$, $\mathbf{B} = \frac{1}{n} \sum_{i=1}^n \mathbf{B}_i$ and $\mathbf{C} = \frac{1}{n} \sum_{i=1}^n \mathbf{C}_i$ and let $a = \frac{1}{n} \sum_{i=1}^n a_i$ and $c = \frac{1}{n} \sum_{i=1}^n c_i$.

Firstly, note that the stochastic Hamiltonian function of (13) has the following form:

$$\mathcal{H}(x) = \frac{1}{2} x^\top \mathbf{Q} x + q^\top x + \ell$$

where $\mathbf{Q} = \begin{pmatrix} \mathbf{A}^2 + \mathbf{B}\mathbf{B}^\top & \mathbf{A}\mathbf{B} - \mathbf{B}\mathbf{C} \\ \mathbf{B}^\top \mathbf{A} - \mathbf{C}\mathbf{B}^\top & \mathbf{C}^2 + \mathbf{B}^\top \mathbf{B} \end{pmatrix} = \mathbf{J}^\top \mathbf{J}$, $q = \begin{pmatrix} \mathbf{A}a - \mathbf{B}c \\ \mathbf{B}^\top a + \mathbf{C}c \end{pmatrix}$, and $\ell = \frac{1}{2}(\|a\|^2 + \|b\|^2)$

The Hamiltonian is thus a smooth quadratic function and the matrix $\mathbf{Q}$ is symmetric and positive semi-definite. In addition, as we assumed that there was at least one solution $x^*$ for the game, i.e. $\xi(x^*) = 0$, we have that $x^*$ is also a global minimum of the Hamiltonian function $\mathcal{H}(x)$, and thus we have that $q = -\mathbf{Q}x^* = -\mathbf{J}^\top \mathbf{J} x^*$. Using this, we can rewrite the Hamiltonian as:

$$\mathcal{H}(x) = \phi(\mathbf{J}x),$$

where the function $\phi(y) := \frac{1}{2} \|y\|^2 - (\mathbf{J}x^*)^\top y + \ell$ is 1-strongly convex with 1-Lipschitz continuous gradient.

Thus, using Lemma D.1 in Loizou et al. [2020], we have that the the Hamiltonian function is a $L_\mathcal{H}$-smooth, $\mu_\mathcal{H}$-quasi-strongly convex function with constants $L_\mathcal{H} = \lambda_{\max}(\mathbf{J}^\top \mathbf{J}) = \lambda_{\max}(\mathbf{Q}) = \sigma_{\max}^2(\mathbf{J})$ and $\mu_\mathcal{H} = \sigma_{\min}^2(\mathbf{J}) = \lambda_{\min}^+(\mathbf{Q})$. $\qquad\square$

### C.2 Experimental Details

We describe here in more details the exact settings we use for evaluating the different algorithms. As mentioned in Section 6, we evaluate the different algorithms on the class of quadratic games:

$$\min_{x_1 \in \mathbb{R}^d} \max_{x_2 \in \mathbb{R}^p} \frac{1}{n} \sum_i \frac{1}{2} x_1^\top \mathbf{A}_i x_1 + x_1^\top \mathbf{B}_i x_2 - \frac{1}{2} x_2^\top \mathbf{C}_i x_2 + a_i^\top x_1 - c_i^\top x_2$$

In all our experiments we choose $d = p = 100$ and $n = 100$. To sample the matrices $\mathbf{A}_i$ (resp. $\mathbf{C}_i$) we first generate a random orthogonal matrix $\mathbf{Q}_i$ (resp. $\mathbf{Q}_i'$), we then sample a random diagonal matrix $\mathbf{D}_i$ (resp. $\mathbf{D}_i'$) where the elements on the diagonal are sampled uniformly in $[\mu_A, L_A]$ (resp. $[\mu_C, L_C]$), such that at least one of the matrices has a minimum eigenvalue equal to $\mu_A$ (resp. $\mu_C$) and one matrix has a maximum eigenvalue equal to $L_A$ (resp. $L_B$). Finally we construct the matrices by computing $\mathbf{A}_i = \mathbf{Q}_i \mathbf{D}_i \mathbf{Q}_i^\top$ (resp. $\mathbf{C}_i = \mathbf{Q}_i' \mathbf{D}_i' \mathbf{Q}_i'^\top$). This ensures that the matrices $\mathbf{A}_i$ and $\mathbf{C}_i$ for all $i \in [n]$, are symmetric and positive definite. We sample the matrices $\mathbf{B}_i$ in a similar fashion with the diagonal matrix $\mathbf{D}_i$ to lie between $[\mu_B, L_B]$[9]. In all our experiments we choose $\mu_A = \mu_C$ and $L_A = L_C$. By varying the different constants $\mu_A, L_A, \mu_C, L_C, \mu_B, L_B$ we can get a variety of games with different properties $\mu, \ell_\xi, \mu_\mathcal{H}, \mathcal{L}_\mathcal{H}$. The bias terms $a_i, c_i$ are sampled from a normal distribution. For further details, see also our source code[10].

---

[9]We highlight that matrices $\mathbf{B}_i$ are not necessarily symmetric.

[10]https://github.com/hugobb/StochasticGamesOpt

As we have already mentioned in Section 6, we pick the step-sizes for the different methods according to our theoretical findings. That is, for constant step-size, we select $\alpha = \frac{1}{2\ell_\xi}$ for SGDA (Theorem 4.1), $\alpha = \frac{1}{4\ell_\xi}, \gamma = \frac{1}{4\mathcal{L}_\mathcal{H}}$ for SCO (Theorem 5.1), and $\gamma = \frac{1}{2\mathcal{L}_\mathcal{H}}$ for SHGD (Corollary 5.3). For the stepsize-switching rule that guarantees convergence to $x^*$, we use the step-sizes proposed in Theorem 4.3 for SGDA and Theorem 5.4 for SCO.

In the experiments, we run all methods (SGDA, SCO and SHGD) using uniform single-element sampling. That is, $|S| = 1$, according to the Definition 2.1. Thus, $\mathbb{P}[i \in S] = p_i = \frac{1}{n}$. By Proposition 3.5, this means that $\ell_\xi = \ell_{\max} = \max\{\ell_i\}_{i=1}^n$. In addition, by Proposition 3.8 in Gower et al. [2019] and the structure of the stochastic Hamiltonian function, $\mathcal{H}(x) = \frac{1}{2}\|\xi(x)\|^2 = \frac{1}{n}\sum_{i=1}^n \frac{1}{n}\sum_{j=1}^n \frac{1}{2}\langle \xi_i(x), \xi_j(x) \rangle$, we have that $\mathcal{L}_\mathcal{H} = \max\{L_{\mathcal{H}_{i,j}}\}_{i=1,j=1}^n$, where $L_{\mathcal{H}_{i,j}}$ is the smoothness parameter of $\mathcal{H}_{i,j} = \frac{1}{2}\langle \xi_i(x), \xi_j(x) \rangle$.

The values of the co-coercive parameters $\ell_i$ for all $i \in [n]$ are computed using $\frac{1}{\ell_i} = \min_{\lambda \in Sp(\mathbf{J}_i)} \Re(\frac{1}{\lambda})$ [Azizian et al., 2020]. Here $Sp(\mathbf{J}_i)$ denotes the spectrum of the Jacobian matrix $J_i$ for all $i \in [n]$ and $\Re$ denotes the real part of a complex number.

### C.3 Additional Experiment: Influence of the Step-size on Convergence

In this section we provide further experiments exploring the performance of SGDA, SHGD and SCO for different constant step-sizes (see Fig. 3). This experiment aims to understand better how the convergence rate and the size of the neighborhood the algorithms converge to depend on the step size. This also enables us to assess if the optimal step-size suggested by the theory is tight. When the step-size is too large we observe that the methods diverges, we do not include them in the plots but provide them below for completeness. We observe that SHGD diverges with a step-size of $\gamma = 3\gamma^*$ with $\gamma^* = \frac{1}{2\mathcal{L}_\mathcal{H}}$, SGDA diverges with a step-size of $\alpha = 4\alpha^*$ with $\alpha^* = \frac{1}{2\ell_\xi}$, SCO diverges with a step-size of $\alpha = 4\alpha^*$ and $\gamma = 4\gamma^*$ with $\alpha^* = \frac{1}{4\ell_\xi}$ and $\gamma^* = \frac{1}{4\mathcal{L}_\mathcal{H}}$. We also observe that SCO is less sensitive to the choice of $\alpha$ than to the choice of $\gamma$

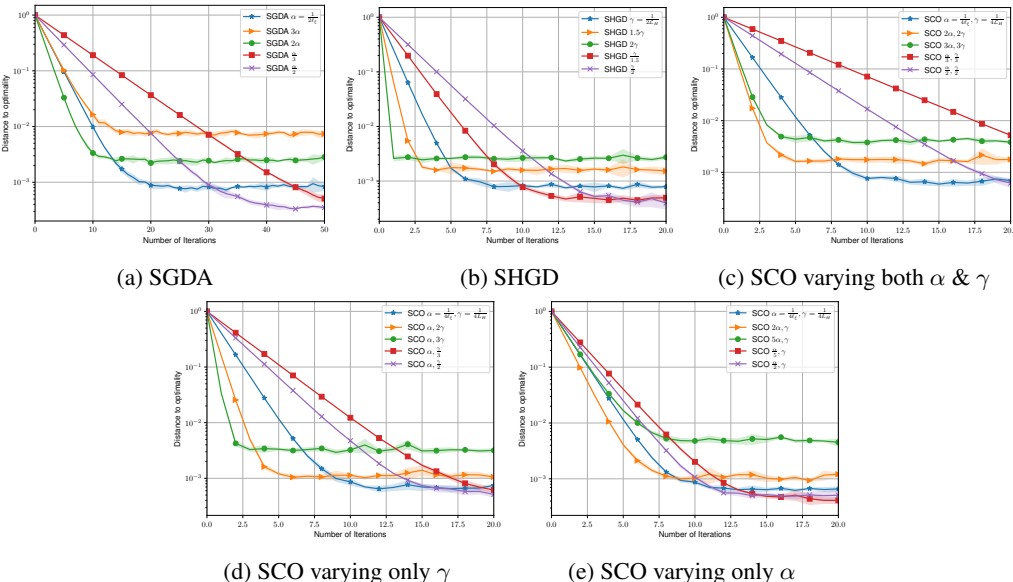

(a) SGDA  (b) SHGD  (c) SCO varying both $\alpha$ & $\gamma$

(d) SCO varying only $\gamma$  (e) SCO varying only $\alpha$

Figure 3: Performance of the methods, SGDA, SHGD and SCO with different constant step-sizes. Distance to optimality: $\frac{\|x^k - x^*\|^2}{\|x^0 - x^*\|^2}$. The step-sizes used are expressed as the optimal theoretical step-size predicted by the theory times a constant. The step-sizes for which the methods diverge are not included in the plots.

# D  Beyond Finite-sum Structure

We explain in this section how all our convergence results also hold for the more general (non finite-sum) stochastic approximation setting [Nemirovski et al., 2009], where we define $\xi : \mathbb{R}^d \to \mathbb{R}^d$ as:

$$\xi(x) := \mathbb{E}_U \tilde{\xi}(x, U), \tag{85}$$

where $U$ is a random vector in $\mathbb{R}^p$, and $\tilde{\xi} : \mathbb{R}^d \times \mathbb{R}^p \to \mathbb{R}^d$ is used to define $\xi$ and is assumed to be well-behaved enough so that (85) exists for all $x$. Setting $U$ to be a uniform random variable in $\{1, \ldots, n\}$ gives back the finite sum setting with uniform weights which was covered in the paper in (1).

We present here only results for the singleton sampling regime, as generalizing to mini-batching and other sampling schemes in the continuous regime is non-trivial and beyond the scope of this paper. This means that we restrict here our estimator for $\xi(x)$ to simply be $\tilde{\xi}(x, U)$, with $U$ sampled according to its distribution. Any appearance of $\xi_v$ in the paper can then be replaced with $\tilde{\xi}(x, U)$ to be able to re-interpret the algorithms and proofs in this setting. In particular, we have that our estimator is $g(x) = \xi_v(x) := \tilde{\xi}(x, U)$, and so $\mathbb{E}[g(x)] = \mathbb{E}_U \tilde{\xi}(x, U) = \xi(x)$ trivially. Also, the expected co-coercivity of $\xi$ is defined as the existence of a $\ell_\xi > 0$ such that

$$\mathbb{E}_U \left[ \|\tilde{\xi}(x, U) - \tilde{\xi}(x^*, U)\|^2 \right] \leq \ell_\xi \langle \xi(x), x - x^* \rangle \quad \forall x \in \mathbb{R}^d. \tag{86}$$

An important quantity appearing in our convergence results is

$$\sigma^2 := \mathbb{E}_U \|\tilde{\xi}(x, U)\|^2, \tag{87}$$

and we assume that $U$ and $\tilde{\xi}$ are such that $\sigma^2 < \infty$.[11] Similarly, $\sigma_{\mathcal{H}}^2$ is defined by letting $u$ and $v$ represent independent singleton sampling in (10), and is assumed to be finite.

Under the assumption of expected co-coercivity of $\xi$ and that $\sigma^2$ is finite, all the convergence theorems of Section 4 and 5 hold as is for $\xi$ more generally defined by (85). This is because all the convergence proofs did not use the finite sum structure explicitly, only the linearity of the expectation operator.

Finally, we can easily generalize Proposition 3.5 for $b = 1$ with $\ell_\xi = \ell_{\max}$ (we assume each $\tilde{\xi}(x, u)$ is $\ell_u$-co-coercive in $x$ around $x^*$, and assume that $\ell_{\max} := \sup_u \ell_u$ is finite).

We can also generalize Proposition 3.6 (for singleton sampling) with $\ell_\xi := \mathbb{E}_U L_U^2 / \mu$ (we assume each $\tilde{\xi}(x, u)$ is $L_u$-Lipschitz continuous in $x$, for each $u$; and that $\mathbb{E}_U L_U^2$ is finite).

# E  More Related Work

The references necessary to motivate our work and connect it to the most relevant literature is included in the appropriate sections of the main body of the paper. Here we present a broader view of the literature, including some more references to papers of the area that are not directly related with our work.

**Smooth monotone games.**  The deterministic version of the problem has been studied extensively both in past and recent work, initially focusing on strongly monotone problems [Tseng, 1995, Gidel et al., 2018, Liang and Stokes, 2019, Azizian et al., 2020, Zhou et al., 2021]. Mokhtari et al. [2020] recently gave rates for extragradient and optimistic gradient through a proximal point approach.

For monotone problems, in the absence of strong monotonicity but assuming a lower bound of the singular values of the coupling between players, Azizian et al. [2020] produce tight results for the extragradient and optimistic gradient methods.

For general monotone problems, Mertikopoulos and Zhou [2019] establish last-iterate convergence, but requires a decreasing step-size schedule; it also does not guarantee convergence to fixed points of non-strictly monotone problems like bilinears. Golowich et al. [2020b] show that last-iterate methods are slower than averaging in this setting. Overcoming the above limitations, Golowich et al. [2020b,a] establish tight upper bounds, $O(1/\sqrt{T})$, for general monotone problems under a weak smoothness assumption for extra-gradient and optimistic gradient respectively.

---

[11]In the finite sum setting, this is always true. But for general random variable $U$, this is not always the case.

**Stochastic smooth monotone games.** Contrary to the deterministic family of problems, in the stochastic setting (where available gradients are noisy), progress has occurred mostly in the past five years. Without variance reduction, some work establishes last-iterate convergence results with a slow (sublinear) rate [Rosasco et al., 2014, Mishchenko et al., 2020]. For *pseudomonotone problems*, Kannan and Shanbhag [2019] give last-iterate convergence of stochastic extragradient with decreasing step sizes, assuming a uniform bound on the variance. Using variance reduction techniques, Palaniappan and Bach [2016] yield fast linear rates without requiring a bound on the variance, however the step size and inner loop length need to be tuned using the modulus of local strong monotonicity. This limitation was later lifted with a variance reduced extragradient method proposed by Chavdarova et al. [2019], which is adaptive to local strong monotonicity.

For monotone problems, in the absence of strong monotonicity but assuming a lower bound of the singular values of the coupling between players, Chavdarova et al. [2019] show that stochastic extragradient with constant step-size does not always yield last-iterate convergence. Such a result for extragradient was shown to be possible using a *double step-size scheme* in Hsieh et al. [2020], using one step size for the extrapolation step and a different step size for the update step.

For general monotone problems, Mertikopoulos and Zhou [2019] use a dual averaging approach to get last-iterate convergence for no-regret algorithms making a bounded variance assumption. This bounded variance assumption is lifted in Lin et al. [2020b]; the authors give finite-time last-iterate convergence of optimistic gradient under other strong conditions: i) for a *relative random noise* model, the noise variance is assumed to be proportional to the squared norm of the vector of gradients, according to the sequence of multiplicative coefficients, $\tau_k$. That is, $\mathbb{E}_i \|\xi_i(x_k) - \xi(x_k)\|^2 < \tau_k \|\xi(x_k)\|^2$. This kind of condition with a constant $\tau_k = \tau$ can occur in some machine learning problems, in particular in supervised learning learning problems when the model is overparametrized and capable to perfectly fitting all training points (so the gradient variance becomes zero at the stationary point of the solution). On the other hand, this assumption is satisfied rarely for adversarial formulations. Even more, Theorem 4.6 in Lin et al. [2020b] requires that the sequence, $\tau_k$, goes to zero with $k$ in order to get convergence; ii) for an *absolute random noise* model, noise variance is uniformly bounded by $\sigma_k$, a bounded variance assumption. Again, for the result of Theorem A.4 in Lin et al. [2020b] to hold, the sequence $\sigma_k$ is assumed to go to zero with $k$. To the best of our knowledge, both assumptions are very strong: in order to get either $\tau_k$ or $\sigma_k$ to decay, one will have to use a mini-batch size that grows up to $n$, which is impractical. It should be noted that the same work contains almost sure (non finite-time) last-iterate results without the above restrictive assumptions.

**Structured non-monotone problems.** The recent works of Daskalakis et al. [2021] and Diakonikolas et al. [2021] show that, for general smooth objectives, the computation of even approximate first-order locally optimal min-max solutions is intractable, motivating the identification of structural assumptions on the objective function for which these intractability barriers can be bypassed.

In this work we focus on a setting (structured non-monotone operators) that one can provide tight convergence guarantees and avoid the standard issues (cycling and divergence of the methods) appearing in the more general non-monotone regime, like . That is, problems satisfying quasi-strong monotonicity (2) or the variational stability condition.

The two classes of VI problems we consider, quasi-strongly monotone and problems satisfying the variational stability condition, have already been used in several papers (under different names). For example, Song et al. [2020] also focuses on quasi-strongly monotone VI but they called them strong coherent VI and show that quasi strong monotonicity is weaker than the strongly pseudo-monotone [Kannan and Shanbhag, 2019] and strongly monotone assumption. In addition, as presented in Song et al. [2020], the non-monotone subsets of quasi-strong monotonicity (a.k.a., strongly pseudomonotone), have many real applications in competitive exchange economy [Brighi and John, 2002], fractional programming [Elizarov and Kalimullina, 2009, Rousseau et al., 2005], and product pricing [Choi et al., 1990]. Meanwhile, the restriction of quasi-strong monotonicity in minimization problems such as one-point convexity [Li and Yuan, 2017] is also used in analyzing neural networks.

In addition in Zhou et al. [2017], the strongly coherent optimization problems have been studied, which are special cases of the quasi-strongly monotone VI. We refer the interested reader to Zhou et al. [2017] for a list of examples of strongly coherent optimization problems. As we mentioned in the main paper, the assumption of quasi-strong monotonicity is equivalent to the concept of strong stability described in Mertikopoulos and Zhou [2019].