# OpenReview forum: "Stochastic Gradient Descent-Ascent and Consensus Optimization for Smooth Games: Convergence Analysis under Expected Co-coercivity"
_NeurIPS.cc/2021/Conference — NeurIPS 2021 Poster_

### Official Review · Reviewer_LrZq · 2021-07-03

**Rating:** 8
**Confidence:** 4

**Summary:**

This paper makes multiple contributions to the convergence analysis of unconstrained stochastic variational inequality problem:

(1) This paper proposes the expected co-coercivity assumption for operators that is equivalent to expected smoothness for functions, and weaker than many prevalent assumptions for variational inequality problem such as bounded gradient, bounded variance and growth condition.

(2) This paper generalizes the unbiased stochastic gradient with mini-batch into unbiased stochastic gradient with continuous spectrum, and applies it to generalize the two prevalent algorithms for unconstrained stochastic variational inequality problem, namely, stochastic gradient descent ascent analysis (SGDA) and Consensus Optimization (CO). The latter is further generalized into Stochastic Consensus Optimization (SCO) by applying to variational inequality problem of finite-sum structure, which contains Combines and generalizes several related algorithms including SGDA, deterministic CO and stochastic Hamiltonian gradient descent (SHGD) as special cases.

(3) This paper gives the first last-iterate convergence analysis of this generalized SGDA that only relies on the expected co-coercivity assumption and the quasi-strongly monotone assumption that are weaker than the previous analysis of SGDA, and recovers the results of deterministic GDA. Effects of hyper-parameters are well studied including constant step-size, diminishing step-size and the optimal batch-size.

(4) This paper gives the first last-iterate convergence analysis of SCO that recovers the results of SGDA, deterministic CO and stochastic Hamiltonian gradient descent (SHGD), and the convergence is even faster than an existing analysis of deterministic CO.

**Ethical Concerns:**

I think this work has no ethical concerns.

**Ethics Review Area:**

["I don’t know"]

**Limitations And Societal Impact:**

Section 7 describes the limitation that quasi strongly monotone assumption could be removed in future work.
Both the authors and I do not think this theoretical work could have any immediate societal impact.

**Main Review:**

Originality: This work makes innovation in both algorithm, assumptions and convergence analysis for unconstrained stochastic variational inequality problem.

Quality: The submission is technically sound. The convergence results are well supported by both theoretical analysis and experimental results. Section 7 describes the limitation that quasi strongly monotone assumption could be removed in future work.

Clarity: This work is clear and well organized.
I appreciate that the relation among EC and other existing assumptions is very clearly shown in a graph.

Significance:
This work has many important innovative results as mentioned in my summary.


Some questions and advice:

(1) At the end of page 3, you said “Thus, a function is L-co-coercive”. Do you mean the gradient is L-co-coercive?

(2) Is x* the global solution in Assumption 3.3?

(3) Typo:  “convergence” could be changed to “converge” in the following two sentences:
(a) At the beginning of Section 3, “It was used to analyze the celebrated forward-backward algorithm (a.k.a, proximal gradient) [29, 8, 37] that is known not to convergence for general monotone operators [5]. ”
(b) At the beginning of Section 5, “[26] show first that CO convergences linearly in the bilinear case. ”

(4) Typo in Theorem 4.3: Change “for $4\lceil\mathcal{K}\rceil$” into “for $k \ge 4\lceil\mathcal{K}\rceil$.”

(5) How does sampling distribution $\mathcal{D}$ or simply batch-size affect the convergence of SCO? I think it better to add this, just as the analysis of the optimal batch-size for SGDA.

(6) Typo right after Theorem 5.1: In the following equation, change $\mu$ into $\mu_{\mathcal{H}}$.
$$\mathbb{E}[||x^{k}-x^*||^{2}] \le(1-\gamma \mu)^{k}||x^{0}-x^{*}||^{2}+\frac{4[\alpha^{2} \sigma^{2}+\gamma^{2} \sigma_{\mathcal{H}}^{2}]}{\gamma \mu}$$

(7) In Theorem 5.4, can the assumptions that $\xi$ is quasi-strongly monotone and expected co-coercive imply that $\mathcal{H}$ is quasi-strongly monotone and expected smooth?

(8) In Figure 2, both subfigures correspond to the same objective function and the difference in $\ell_{\xi}$ is caused by the hyper-parameter choice difference, right? If so, I think the 4 curves can be plotted in 1 figure.

**Time Spent Reviewing:**

About 3 hours

---

> ### Author Response · Authors · 2021-08-10
> **Response to Reviewer LrZq**
>
> **Thanks!**
>
> Thank you for the positive review and for acknowledging the multiple contributions to the convergence analysis of algorithms for solving the unconstrained stochastic variational inequality problem.
> Thank you also for highlighting the originality and quality of our results and for emphasizing the significance of our work.
>
> **Response to all questions raised**
>
>  Please find below our response to all questions raised.
>
> * Point (1). At the end of page 3, yes we mean “gradient is L-co-coercive”.
>
> * Point (2). In Assumption 3.3, $x^*$ is indeed the global solution as defined in the first page of the paper (line 21).
>
> * Points (3+4+6). In the camera-ready version we will fix all typos as you suggested. Thanks for catching these typos.
>
> * Point (5). Following a similar approach to the one we use for the SGDA one can obtain interesting results on the selection of distribution $\cal D$ for the SCO. We can definitely add more details on this in the updated version of the paper and explain what is the closed-form expression for the optimal minibatch size for SCO and how it is related to the SGDA result. This could be obtained as a corollary of Theorem 5.1. We will elaborate in the camera ready version of the paper as there will be more space there. This is a great suggestion, thanks!
>
> * Point (7). In Theorem 5.4 the assumptions that $\xi$ is quasi-strongly monotone and expected co-coercive do not necessarily imply that the Hamiltonian function $\cal H$ is quasi-strongly monotone and expected smooth. However if the game is strongly-monotone quadratic then this is true as we show in our numerical experiments and in Appendix C.1.
>
> * Point (8). In the camera-ready version we can update Figure 2 as it suggested.

---

> > ### Comment · Reviewer_LrZq · 2021-08-23
> > **Reviewer LrZq’s 2nd response**
> >
> > The authors' response well addresses my concern.
> >
> > I also read the other reviewers' opinion and the authors' corresponding response, and I accordingly have new question and suggestion:
> >
> > (1) In the authors' response to reviewer kKUx, the authors admit that EC implies co-coercity around $x^*$. If we weaken EC to co-coercity around $x^*$, can we still prove the convergence?
> >
> > (2) In Line 317, based on the author's response to reviewer kKUx, I think it would be better to say:
> > "We provided the first convergence analysis of SGDA and SCO without the strong bounded noise assumption by introducing the much weaker expected co-coercivity assumption." Is this convergence rate or complexity more efficient than previous SGDA analysis? If the same, then I think it better to remove "efficient" and highlight the advantage of weaker assumption.
> >
> > After reading, I still feel that this paper is sufficiently novel for Neurips, in terms of both assumption and proving technique (to extend the proving technique of minimization problem to VI is non-trivial), though the limitations about the quasi-strongly monotonicity (compared with [28]) and assumption of unique solution mentioned by reviewer 94Qg are correct. Hence, **I keep my rating=8.**

---

> > > ### Author Response · Authors · 2021-09-01
> > > **Response to Reviewer LrZq**
> > >
> > > Point (1). In our current proofs, the EC is required, as it allows us to obtain an upper bound for the $\mathbb{E}_{\cal D}||\xi_v(x)||^2$, (see Lemma 3.4). For this reason, we believe that EC is the natural notion for the stochastic setting. \
> > > In the deterministic setting (distribution $\cal D$ corresponds to the full-batch sampling), assuming co-coercivity around $x^*$ is sufficient, and in this case, we can prove convergence. Note that the proposed theory captures this as special case.
> > >
> > > Point (2).  Thank you for the suggestion. We will update that sentence in the camera-ready version to correspond exactly to our main contributions. That is, (i) first convergence analysis for SCO and (ii) convergence analysis of SGDA under weaker conditions.
> > >
> > > Thank you again for the discussion and the feedback on our work.

---

> > > > ### Comment · Reviewer_LrZq · 2021-09-01
> > > > **Reviewer LrZq’s 3rd response**
> > > >
> > > > OK. Great.

---

### Official Review · Reviewer_94Qg · 2021-07-15

**Rating:** 7
**Confidence:** 5

**Summary:**

This paper introduces the expected co-coercivity condition, explain its benefits, and provide the first-iterate convergence guarantees of stochastic gradient descent ascent (SGDA) and stochastic consensus optimization (SCO) under this condition for solving a class of stochastic variational inequality problems that are potentially non-monotone. The authors prove linear convergence of both methods to a neighborhood of the solution when the algorithms use constant step-size, and propose insightful stepsize-switching rules to guarantee convergence to the exact solution. One of the appealing features is that the convergence guarantees hold under the arbitrary sampling paradigm, and as such, the authors give insights into the complexity of mini-batching.

**Limitations And Societal Impact:**

1. Compare their conditions on noises to the existing assumptions in the literature more carefully.

2. Elaborate more on the role of quasi-strong monotonicity in their proof and compare their results to [28].

3. Explain why the existence of the solution will not be an issue and comment on practical implications of their results.

**Main Review:**

My overall opinion on the manuscript is positive. The motivation of expected co-coercive condition is strong and the results in Sections 4 and 5 on the convergence analysis of SGDA and SCO are certainly nontrivial. The combination of the finite-sum structure and the unbiased estimators proposed in [30] seems to be intriguing.

Nevertheless, this paper seems like a weak accept to me. There are a number of new and interesting results in non-trivial settings but it seems that the authors overclaim their contributions. To be more specific, I encourage the authors to consider the following points:

First, the authors compare their assumptions to the standard assumptions in the literature (cf. Table 1) and demonstrate the advantage of the proposed EC condition. While I agree that EC condition is new and useful for analyzing the algorithms, a few assumptions, e.g., bounded gradient condition, seem necessary in a more general non-smooth setting. It is somehow too much to say that expected smoothness/expected cocercivity is one of the weakest assumption for analyzing SGD since they only complement the existing conditions in my mind.

Second, I bring your attention to the following article:

P. Mertikopoulos and Z. Zhou. "Learning in games with continuous action sets and unknown payoff functions". Mathematical Programming. 173: 465–507. 2019.

In Definition 6.1, the notion of strong stability is defined and resembles the quasi-strong monotonicity in Eq. (2) of the current paper. To the best of my knowledge, such notion has not been adopted to analyze SGDA and SCO so the convergence results in the current manuscript are new. But the authors might cite the above reference and give some credits.

Third, I think it is a bit unfair to compare your results with [28] and claim that the assumptions on $\mathbb{E}[\|\xi_i(x)\^2|]$ are unrealistic (Agreed. The assumptions are very restrictive but seem necessary in the setting of [28]). More specifically, all the last-iterate convergence results in this paper are proved under the quasi-strongly monotonicity, which leads to a contraction (even with constraint sets) and plays the similar role as the strong monotonicity from a technical point of view. So there is no need to impose further conditions on the noises. However, the last-iterate convergence analysis in [28] is done without the quasi-strong monotonicity. I check the proof details in the supplementary material and believe that the quasi-strongly monotonicity can not be removed. Then, it gives rise to the question on which one is more realistic, the quasi-strongly monotonicity or the assumption on $\mathbb{E}[\|\xi_i(x)\^2|]$? The current writing is somehow misleading in the sense that your theorems strictly improve those ones in [28] without imposing any additional assumptions. It would be better to elaborate more on this point and precisely position the contribution of your paper in the literature.

Finally, if the quasi-strong monotonicity holds true, I am curious whether the solution must be unique (if exist) or not. Moreover, in the unconstrained setting without the quasi-strong monotonicity, the existence of the solution will be an issue. Therefore, I encourage the authors to elaborate more on it, justifying why their results have practical implications (note that most of the real application problems are in constrained setting).

**Time Spent Reviewing:**

4 hours

---

> ### Author Response · Authors · 2021-08-10
> **Response to Reviewer 94Qg**
>
> ## **Thanks**
> Thank you for the overall positive opinion on our work and for highlighting the strong motivation of expected co-coercivity. Thank you also for acknowledging the importance of our results on the convergence analysis of SGDA and SCO and praising their novelty.
>
> ## **Response to all questions raised**
>
> **[On Expected Co-coercivity].**
> Our work focuses on smooth games as indicated by the title. However even in this setting existing results required strong assumptions like bounded gradient and bounded variance. It is under this setting that we claim that EC is one of the weakest assumptions for analyzing SGDA and SCO (and one of the weakest for analyzing stochastic algorithms for solving unconstrained VI problems). We agree with Rev. 94Qg, that for the more general non-smooth setting condition like the bounded gradient is necessary. We can be more precise and easily clarify that in the updated version of our work.
>
> However we highlight that it is possible for an operator to be $\ell$-co-coercive around $x^*$ and not be Lipschitz continuous. This highlights the wider applicability of the $\ell$-co-coercivity around $x^*$ assumption that is all we need for several of our convergence results, in contrast to the Lipschitz continuity of $\xi$ which is typically assumed in the variational inequality literature. In Appendix A.6, we provide such example of a $\mu$-quasi strongly monotone operator that is `$\ell$-co-coercive around $x^*$ , which is not monotone nor Lipschitz continuous.
>
> **[Give credits to work [32] ].**
> We note that we have already cited the suggested work as reference [32]. We will make sure to add a comment on the connection between the quasi-strong monotonicity of our paper to the concept of strong stability presented in [32]. We also highlight that the “variational stability condition” is the same in both papers (equation 2.10 in [32] and line 22 in our work). Thank you for the pointer!
>
>
> **[Comparison to the results of [28] ].**
> Thank you for raising this issue. We agree with the comments of Reviewer 94Qg on the theoretical results of paper [28] and we will make sure that we will clarify all points in the camera-ready version of our work. In the current comparison we focus only on the noise assumptions, and we mention that [28] requires much stronger noise conditions. However, as Rev.94Qg pointed out, [28] provides last iterate convergence guarantees for SGDA without assuming a strong-monotonicity type of assumption which itself is quite impressive and should be highlighted in the updated version.
>
> In the current version we write: *``[28] requires a bound on the variance with vanishing constants, which as far as we know, can only be satisfied by running the algorithms with growing mini-batch size”*. The growth condition bound of [28] is more restrictive than our setting (in terms of noise assumption), however by having it the authors were able to provide last iterate convergence guarantees of SGDA without the quasi-strong monotonicity. We will add a clarification comment at this point in the updated version of our work.
>
> Regarding the comment  *“Then, it gives rise to the question on which one is more realistic, the quasi-strongly monotonicity or the assumption on $\mathbb{E}_{\cal D}[||\xi_i(x)||^2]$ "* we can add more clarification points in the paper. If for example, the operator $\xi$ turns out to be strongly monotone then quasi-strong monotonicity is satisfied for free, while the growth condition of [28] will be restrictive and might not be satisfied if the algorithm is run with uniform single element sampling (not increasing mini-batch size). On the other hand, there might also be examples where the operator $\xi$ is not quasi-strongly monotone, and in order to guarantee last iterate convergence one need to use the growth condition from [28].
>
> Finally, we note that we have already devoted a full paragraph in Appendix E on the results presented in [32] and [28]. We will make sure to include this comparison in the main paper of the camera-ready version as there will be more space there. In particular, we are planning to include this paragraph in the “Connection to other Assumptions” subsection of the main paper.
>
> Having a thorough comparison between the two approaches and assumptions used is actually a very good suggestion and will help the reader to better understand where our contributions belong  in the literature, thanks!
>
> In the last section of our work we mentioned that one of our future goals is to provide last-iterate convergence guarantees of methods for solving stochastic VI problems without the quasi-strong monotonicity. We speculate that a combination of EC with the noise assumptions proposed in [28] could potentially make that possible.
>
> **[On Quasi-strong monotonicity and unique solution].**
> Quasi-strong monotonicity does not necessarily require a unique solution. As we mentioned in a footnote in the first page of the paper, assuming a unique solution can be relaxed but for simplicity of exposition we enforce it in our work.
> If the quasi-strong monotonicity holds (without assuming unique solution) we can have a convex set of solutions (to show this we just need to consider two solutions and the quasi-strong monotonicity condition at their convex combination that is not a solution to obtain a contradiction) but the $x^*$ considered in the quasi-strong monotonicity should be the projection of $x$ onto the solution set. An example of such a  quasi-strong monotone vector field is: $ \xi(x) = x$ if $x<0$; $\xi(x) = 0$ if $0\leq x \leq 1$; and $\xi(x) = x-1$ if $x>1$. Quasi strong monotonicity should be seen as a generalization of quasi-strong convexity appearing in optimization literature (see Definition B.3 and papers [55, 57] ).
>
> In the unconstrained setting, the existence of a solution is an issue in general. Just like in unconstrained minimization the objective may not be lower-bounded, in unconstrained games there may be no solution to the VIP problem (even when assuming monotonicity). Studying the sufficient conditions for the existence of a solution for our problem is beyond the scope of this paper. There are several works that study solution concepts for games in the context of machine learning [1,2,3].  We can definitely elaborate more in the updated version and add comments to this end.
>
> * [1] Jin, Chi, Praneeth Netrapalli, and Michael Jordan. "What is local optimality in nonconvex-nonconcave minimax optimization?." International Conference on Machine Learning. PMLR, 2020.
> * [2] Gidel, Gauthier, et al. "A Limited-Capacity Minimax Theorem for Non-Convex Games or: How I Learned to Stop Worrying about Mixed-Nash and Love Neural Nets." AISTATS (2021).
> * [3] Zhang, Guojun, Pascal Poupart, and Yaoliang Yu. "Optimality and Stability in Non-convex Smooth Games." arXiv preprint arXiv:2002.11875 (2020).
>
> ## **Final Comment to Reviewer 94Qg**
> We believe that all points raised by reviewer 94Qg can be easily handled in the updated version of our work. As we mentioned in our response we can provide more details on the expected co-coercivity and how is connected with other existing conditions, have a thorough comparison of our results compare to papers [32] and [28] and add comments on the quasi-strong monotonicity and uniqueness/existence of the solution.
>
> **We believe that all of the above are minor corrections, and we hope that reviewer 94Qg can consider updating their assessment.**

---

> > ### Comment · Reviewer_94Qg · 2021-08-28
> > **Thank you for your rebuttal**
> >
> > The authors' response addresses most of my concerns.
> >
> > Importantly, I see one interesting point in your respond to other reviewers. That is, your convergence guarantees are the first that provide convergence rates of the two algorithms under study (SGDA and SCO) without assuming extra assumptions on the noise or the variance for the classes of problems we explore. In my mind, the quasi-strong monotonicity has been widely used in the literature and the proof techniques do not differ significantly from that used for strong monotone case. However, only assuming the bounded noise of stochastic gradient at optimal solution seems a quite appealing feature, and with such relaxation, I think it is fine to keep quasi-strong monotoncity.
> >
> > Nevertheless, I hope to bring the authors' attention to several sentences in the introduction, e.g., "In this paper, we address both of these issues. We generalize the recent improved analysis for SGD to the case of unconstrained stochastic variational inequality, and prove the last-iterate convergence for both SGDA and SCO without requiring any bounded variance assumption". Unintentionally, you emphasis "without any bounded variance assumption" but do not say "add quasi-strong monotonicity". From a technical point of view, it is not difficult to replace growth condition by bounded condition in [28] if quasi-strong monotonicity is assumed. Also, the last-iterate convergence of stochastic GDA (even with zero noise) can fail in general for specific blinear games. Therefore, I strongly suggest the authors to carefully revise your paper and clearly position their contribution.
> >
> > Only assuming bounded noise at optimal solution is technically interesting to me. Hence, I raise my score to 7.

---

### Official Review · Reviewer_kKUx · 2021-07-16

**Rating:** 6
**Confidence:** 4

**Summary:**

This paper presents a new analysis for solving smooth, unconstrained variational inequalities, particularly focused on applications to minimax optimization. The primary theory innovation in this work is the observation that the very standard cocoercivity assumption can be replaced by cocoercivity holding in expectation. For such problems, convergence guarantees for the Stochastic Gradient Descent-Ascent and Stochastic Consensus Optimization are given. In both cases, O(1/T) convergence rates are proven. Numerics are given showing the proposed stepsize schemes perform reasonably.

**Limitations And Societal Impact:**

The only application thoroughly considered by the work is a simple bilinear matrix game. This may be partly due to the above comment that assuming strongly monotone and cocoercivity at x^* heavily limits the range of applications. Discussing these limitations up front and the limitations like cycling and divergence that are known to occur in non-monotone settings would give a fuller picture of where these methods can be guaranteed to work and the scale of advancement this work represents.

**Main Review:**

The proposed EC condition is a special case of the whole function $\xi(x)$ being cocoercivity at x^* (as an immediate consequence of Jensen's inequality + EC). As a result, all of the considered problem instances are strongly monotone and cocoercivity at x^*. So the geometry around the limit point is assumed to be very convex-concave. Including meaningful families of nonconvex-nonconcave problems satisfying this would greatly help motivate the work.

A few times, the paper is prone to overclaim its novelty/importance. Below are a few key examples:
-Line 146: "There is no analysis using a concept similar to our [EC]". EC is a natural extension of the very standard cocoercivity assumption. The analysis for SGDA does not deviate notably from the standard model of recursive analysis used for strongly monotone and cocoercivity operators. Also, as noted in Table 1, the EC condition is similar to Expected Smoothness.

-Line 168: "Theorem 4.1 is the first last-iterate non-asymptotic convergence guarantee for SGDA..." That sentence then continues to contradict itself pointing out the work of Lin et al. does provide a prior last-iterate non-asymptotic convergence guarantee.

-Line 317: "We provided the first efficient convergence analysis of SGDA and SCO by introducing the expected co-coercivity assumption." I don't know what this claim means, but it is certainly not the case that all prior works have been inefficient.


Minor comments
-Line 283: Appears to be missing identity matrices in the p.s.d. matrix upper and lower bounds.

**Time Spent Reviewing:**

5

---

> ### Author Response · Authors · 2021-08-10
> **Addressing the comment in “Limitations and Societal Impact” section**
>
> **The comment of the reviewer:**
>
>  *“The only application thoroughly considered by the work is a simple bilinear matrix game. This may be partly due to the above comment that assuming strongly monotone and co-coercivity at x^* heavily limits the range of applications. Discussing these limitations up front and the limitations like cycling and divergence that are known to occur in non-monotone settings would give a fuller picture of where these methods can be guaranteed to work and the scale of advancement this work represents.”*
>
> **Our response:**
>
> In terms of applications (in experiments) we focus on simple stochastic strongly monotone quadratic problems, as we wanted to verify the theoretical results which is the main contribution of our work. We note that one can easily construct non-monotone and non-Lipschitz operators that satisfy our assumptions. See for example the proposed operator in Appendix A.6.
>
> In the updated version we can easily discuss the limitations of our assumptions in the introduction and provide more details on the cycling and divergence of the methods that might occur in general non-monotone settings. We agree with the reviewer that this will give a better picture on the setting under study and help the reader understand the scale of advancements of our results. SGDA and SCO are popular algorithms and have already been used in several settings including challenging problems like GANs. In our work we focus on a setting (structured non-monotone operators) that one can provide tight convergence guarantees and avoid the standard issues appearing in the more general non-monotone regime.

---

> ### Author Response · Authors · 2021-08-10
> **On novelty/importance of our results**
>
> Reviewer kKUx, mentions three specific points where it looks like we overclaim the novelty/importance of our results.
>
> We handle and provide more details for each one of these three points below.
>
> * Line 146: *“There is no analysis using a concept similar to our [EC]".*
>
> We still support this claim and we believe that it is not an exaggerated statement. All existing analyses of SGDA for strongly monotone and co-coercive operators require the much stronger extra assumptions of “bounded noise” or “bounded variance” to guarantee convergence. In Appendix A.5 and Proposition A.7 we explicitly prove why the combination of strong monotonicity and bounded noise lead to a contradiction (empty set of operators) in the unconstrained setting. To the best of our knowledge, the only work that provides last-iterate convergence guarantees of SGDA is paper [28] that shows convergence under a growth condition (stronger condition on the noise compared to EC) and in a different setting than ours.  Thus, claiming that “The analysis for SGDA does not deviate notably from the standard model of recursive analysis” is an oversimplification as all existing analyses of SGDA require much stronger assumptions than our theory.
>
> Regarding the SCO, we highlight that there are no known convergence guarantees in the literature, even if the method is used extensively for solving large-scale adversarial problems. All existing results focus on deterministic variants. As such providing the first convergence guarantees of this algorithm is a big deal. And we were able to do that using the EC condition.
>
> The use of EC also allows us to provide novel analysis of the deterministic GDA and Consensus Optimization as special cases of our theorems. All previous analyses were not able to do that as the definition of $\sigma$ (see eq. 6) was not a variance at the optimum and as a result could never be equal to zero. We highlight that we also give the first stochastic reformulation of the variational inequality problem that enables us to provide convergence results for  the arbitrary sampling paradigm and give insights into the complexity of minibatching. Thus, we provide analysis of SGDA and SCO beyond the standard uniform single element sampling.
>
> Finally, regarding the comment *“the EC condition is similar to Expected Smoothness”*, we highlight that the expected smoothness (ES) is not able to be used as it is in the VI setting.  ES includes function suboptimality in the upper bound, a quantity that is not useful in the VI setting. As we explain in our work, EC is the true generalization of ES in the operator setting, and the two notions are equivalent only in convex and smooth single-objective optimization problems.
>
> * Line 168: *"Theorem 4.1 is the first last-iterate non-asymptotic convergence guarantee for SGDA..."*
>
> Reviewer kKUx, mentions that this sentence continues to contradict itself pointing out that the work of Lin et al. does provide a prior last-iterate non-asymptotic convergence guarantee.
> The full statement in our work was the following:
>
> *“To the best of our knowledge, Theorem 4.1 is the first last-iterate non- asymptotic convergence guarantee for SGDA, in addition to the result presented in [28]. However, the convergence result from [28] required much stronger noise conditions. For example, a bound on the variance with vanishing constants, which as far as we know, can only be satisfied by running the algorithms with growing mini-batch size [14] (see also App. E for a more detailed discussion).”*
>
> The above statement is not an overclaim of the novelty of our result. Our results are indeed the first last-iterate non- asymptotic convergence guarantee for SGDA under the setting under study without assuming extra condition on the noise. In particular, our theory provides the first last-iterate linear convergence to a neighborhood (Theorem 4.1, for constant step-size) and $O(1/k)$ convergence to the exact solution (Theorem 4.3 for decreasing step-size) for quasi-strongly monotone operators (a main difference compared to [28] which uses growth conditions). We can easily update the above part in the camera-ready version of our work to correspond to the above discussion.
>
> * Line 317: *"We provided the first efficient convergence analysis of SGDA and SCO by introducing the expected co- coercivity assumption."* I don't know what this claim means, but it is certainly not the case that all prior works have been inefficient.
>
> This is again no exaggeration. It represents exactly the main contribution of our work. We provide the first convergence guarantees of SCO, one of the most successful algorithms for solving large-scale adversarial problems and we were able to do that by introducing/using the expected co-coercivity condition. To the best of our knowledge our work is the first that provides convergence guarantees for SCO.
> Also as we explained above all prior works focusing on the analysis of SGDA for (quasi) strongly monotone operators required unrealistic strong assumptions like bounded noise (which come in contradiction with the strong monotonicity). Using the expected co-coercivity assumption we were able to remove such assumptions and provide meaningful convergence guarantees.
>
> We have clearly argued the value of our work by providing more details. **As we explain, none of the points/sentences mentioned is an overclaim of the importance of our results and we do not think that the exact wording of these statements is reason for suggesting rejection.**  That said, we will gladly update the mentioned statements in the camera-ready version to correspond exactly to the further explanation given above.
>
> ---------------------------------------------------
>
> **[On Minor comments]**
>
> -Line 283: Appears to be missing identity matrices in the p.s.d. matrix upper and lower bounds.
>
> You are right. The identity matrices are missing. Thanks for catching that.

---

> > ### Comment · Reviewer_kKUx · 2021-08-23
> > **Response to authors**
> >
> > I agree with the authors that all of the statements above are true in some senses, but the nuance explained in your above reply is not included in many of these statements in the manuscript.
> >
> > For example, rather than claiming "Theorem 4.1 is the first last-iterate non-asymptotic convergence guarantee for SGDA, in addition to the result presented in [28]", more realistically in my view and as you explain above, the paper improves over existing non-asymptotic convergence guarantees of [28] by replacing assumptions X with Y.
> >
> > As written, I still think the manuscript overclaims the nature of many results (claiming to be the first in many respects in relation to SGDA, when the results are more accurately incrementally relaxing assumptions). I agree that "the exact wording of these statements is [not] reason for suggesting rejection". Although I think the paper would benefit from prioritizing nuance currently missing from its claims, my primary concern with the manuscript is the issues raised above with the model requiring convex-concave like geometry around $x^\star$ and the resulting limits in terms of the claimed nonconvex-nonconcave applications.

---

> > > ### Author Response · Authors · 2021-08-26
> > > **Thanks / All issues handled**
> > >
> > > There were two main issues pointed out by reviewer Kkux in their original review: \
> > > (i) on the setting under study and\
> > > (ii) on the importance/novelty of our result.
> > >
> > > With our response, we provide more details on these topics.
> > >
> > > As the reviewer mentioned in his last message, “the exact wording of these statements is **[not]** reason for suggesting rejection". This handles point (ii), and as we explained, our statements were not exaggerations. We agree with the reviewer that the paper would benefit from prioritizing the nuance currently missing from the claims. According to NeurIPS policy the previous years, we will have an extra page available for new content (incorporating feedback from the reviewers).  In the updated version, we will include exactly all the details mentioned above to clarify further the main contributions of the work.
> > >
> > > Regarding point (i) on the setting under study, we explain in a comment above why the assumptions of our work are not unrealistic and appear in several state-of-the-art recent papers in the area. We note again that in our work, we focus on a setting (structured non-monotone smooth operators) that one can provide tight convergence guarantees and avoid the standard issues (cycling and divergence of the methods)  that might occur in the more general non-monotone regime.
> > >
> > > From the feedback of the reviewer and the above discussion, we hope that we clarify all points raised. If you agree that we managed to address all issues, please consider raising your mark to acceptance.
> > > If you believe this is not the case, please let us know so that we have a chance to respond.
> > >
> > > Thank you again for the discussion and the feedback on our work.

---

> > > > ### Comment · Reviewer_kKUx · 2021-08-29
> > > > **Revised Review**
> > > >
> > > > Thank you for your detailed responses further clarifying my concerns with the paper.
> > > > I am largely satisfied that the presented theory applies reasonably widely (although not to the fullest extend of nonconvex-nonconcave applications discussed early on in the manuscript) and that further nuance can be included when discussing the paper's contributions.
> > > > I am raising my score to be slightly above acceptance.

---

> ### Author Response · Authors · 2021-08-10
> **On setting under study**
>
> We agree with Rev.kKUx that the following holds:
>
> $$ ||\xi(x)-\xi(x^*)||^2=||\mathbb{E}{\cal D} [\xi_v(x)-\xi_v(x^*)] ||^2 \overset{Jensen}{\leq} \mathbb{E}_{\cal D} [ ||\xi_v(x)-\xi_v(x^*)||^2 ] \overset{EC}{\leq}  \ell \langle \xi (x),x-x^*\rangle $$
>
> That is, if EC is satisfied then (by Jensen’s inequality) the operator $\xi$ is co-coercivity around $x^*$.
>
> The comment: "all of the considered problem instances are strongly monotone and co-coercivity at $x^*$" is not accurate. While all of our results on the convergence of SGDA require the $\mu$-quasi strong monotonicity, our Theorems 5.1 and 5.4 on Stochastic Consensus Optimization (SCO) do not require $\xi$ to be quasi-strongly monotone to guarantee convergence. SCO converges even if $\xi$ satisfies the weaker variational stability condition $\langle\xi(x),x-x^*\rangle \geq0$.
>
> Let us also note that while a $\ell$-co-coercive operator is always $\ell$-Lipschitz continuous, it is possible for an operator to be $\ell$-co-coercive around $x^*$ and not be Lipschitz continuous. This highlights the wider applicability of the $\ell$-co-coercivity around $x^*$ assumption, that is all we need for several of our convergence results, in contrast to the Lipschitz continuity of $\xi$ which is typically assumed in the variational inequality literature. In Appendix A.6, we provide such example of a $\mu$-quasi strongly monotone operator that is $\ell$-co-coercive around $x^*$ , which is not monotone nor Lipschitz continuous.
>
> The assumption of quasi-strong monotonicity is equivalent to the concept of strong stability described in [32]. The variational stability condition was also used in several other papers (see for example [21, 54]) or even in popular books like [13] and encompasses a wide range of nonmonotone problems. We can elaborate more on the class of problems satisfying these conditions in the updated version of the paper. In addition, we should highlight that in optimization literature, smooth quasi (strongly) convex problems appeared in several applications. See for example [17] and [57]. Our assumptions are extensions of these settings to monotone operators.
>
> **Remark:** However, the most important point that we should raise is that even if the VI problem has a strongly monotone and Lipschitz continuous operator (which is the most basic setting), our theoretical analysis will still be novel. As we mentioned  in our paper and as will explain below, to the best of our knowledge, there were no known last-iterate convergence guarantees for SGDA for the setting under study without extra assumptions/bounds on the noise of the problem, before our work. In addition, our paper is the first that provides convergence guarantees for SCO. All existing works focus only on the much simpler deterministic setting. The goal of our work is to provide tight convergence guarantees for two of the most prominent algorithms for solving VI problems (SGDA and SCO) in a setting where this is possible. We believe that our work will work as a first step towards a better understanding of the behaviour of these methods in general non-monotone problems (which is the ultimate goal).

---

> > ### Comment · Reviewer_kKUx · 2021-08-23
> > **Response to authors**
> >
> > I agree that the SCO results only require a weaker stability condition than quasi-strong monotonicity.
> >
> > Focusing on the SGDA analysis, I still believe any function satisfying these two conditions will have "the geometry around the limit point assumed to be very convex-concave". For example, taking the author's proposed toy example from Appendix A.6. This operator $\xi(x)$ around the origin has a very positive definite Jacobian:
> > $$ \nabla \xi(0) = L I. $$
> > In the case of applying GDA, such a positive definite jacobian corresponds to being strongly convex-strongly concave near the origin. Under such conditions, it is not very surprising to me then that last-iterate convergence can be achieved. That is not to say that the Theorems in the paper are not new, but rather, I am not convinced of their importance or wide application.
> >
> > Particularly, I feel the comment "Including meaningful families of nonconvex-nonconcave problems satisfying this would greatly help motivate the work." has not been addressed. The work claims as motivation GANs, robust learning, and some forms of reinforcement learning. Showing some nonconvex-nonconcave version of one of these applications meets the assumptions would address my skepticism on the range of real problems that fit under this paper's structured model.

---

> > > ### Author Response · Authors · 2021-08-26
> > > **Further clarifications**
> > >
> > > **Thanks**
> > >
> > > Thank you for the response and for initiating a discussion.
> > >
> > > Let us start by referring Reviewer kKUx, to the bullet points of the main contributions of our work at the beginning of our rebuttal and the comments of Reviewer LrZq, as they greatly describe the significance of our work.
> > >
> > > **Importance of Results / Motivation:**
> > >
> > > Our main motivation in assuming the quasi-strong monotonicity (or the more relaxed variational stability condition) is to prove explicit global last-iterate convergence results for stochastic algorithms solving the stochastic variational inequality problem under conditions as weak as possible.
> > >
> > > We note that, in our opinion, our theoretical results are novel even if we simply assume that the VI problems are strongly monotone (instead of quasi strongly monotone) or monotone (instead of assuming the variational stability condition).
> > >
> > > *On the SGDA analysis:* \
> > >  Saying that under our conditions, it is not very surprising that last-iterate convergence can be achieved for SGDA is an understatement of our contribution. It might not look surprising, but to the best of our knowledge, our convergence guarantees are the first that provide convergence rates of the two algorithms under study (SGDA and SCO) without assuming extra assumptions on the noise or the variance for the classes of problems we explore. In our opinion, this by itself is a significant contribution. We disagree with the characterization of the reviewer (in the next comment) that our analysis of SGDA  “incrementally relaxing assumptions”. As we mentioned in our original response, in Appendix A.5 and Proposition A.7 we explicitly prove why the combination of strong monotonicity and bounded noise leads to a contradiction (empty set of operators) in the unconstrained setting. Thus finding the correct/natural assumptions under which one can guarantee last-iterate convergence was an open problem that we resolved.
> > >
> > > Our analysis of SGDA also captures any meaningful sampling strategy that can be used in the update rule (through the arbitrary sampling paradigm). This was never explored before for stochastic algorithms solving VI problems. In addition the relative random noise assumption used in [28] $(E_{i}||\xi_i(x_k)-\xi(x_k)||^2 <\tau_k ||\xi(x_k)||^2$ with $\tau_k$ be a decreasing sequence) can only be satisfied by running the algorithms with growing mini-batch size. Thus does not really cover the behaviour of the standard SGDA algorithm with fixed minibatch size in the setting under study. Finally the choices of step-sizes proposed in our work were also not considered before.
> > >
> > > *On the setting:* \
> > >  We agree with the reviewer. In the introduction of our paper, we mentioned that min-max problems appear in several machine learning applications including GANs, robust learning and reinforcement  learning. This was precisely our motivation for working on these problems. However, there is no point in the paper that we claim that our theory applies to everything; Our main claim is that we make significant theoretical progress on the convergence guarantees for two or the most prominent algorithms in the literature (SGDA and SCO) that are currently used for solving these challenging problems.
> > >
> > > In practical scenarios, min-max problems like GANs are highly non-monotone and potentially non-smooth. In our work we focus on a setting (structured non-monotone smooth operators) that one can provide tight convergence guarantees and avoid the standard issues appearing in the more general non-monotone regime.
> > >
> > > **More details on class of problems:**
> > >
> > > Reviewer kKUx believes that the fact that our assumptions imply convex-concave like geometry around x^*, is restrictive in terms of applications. We disagree with this statement and we provide our reasoning below.
> > >
> > > At this point we note that recent works of Daskalakis et al. (2021) [1] and Diakonikolas et al. (2021) [2] show that, for general smooth objectives, the computation of even approximate first-order locally optimal min-max solutions is intractable, motivating the identification of structural assumptions on the objective function for which these intractability barriers can be bypassed.
> > >
> > > This is exactly what we did in this work. We focus on a setting (structured non-monotone operators) that one can provide tight convergence guarantees and avoid the standard issues appearing in the more general non-monotone regime.
> > >
> > > The two classes of VI problems we consider, quasi-strongly monotone and problems satisfying the variational stability condition, have already been used in several papers (under different names). For example, Song et. al (2020) [3] also focuses on quasi-strongly monotone VI but they called them strong coherent VI and show that quasi strong monotonicity is weaker than the strongly pseudo-monotone [4] and strongly monotone assumption. In addition, as presented in [3], the non-monotone subsets of quasi-strong monotonicity (a.k.a., strongly pseudomonotone), have many real applications in competitive exchange economy [6], fractional programming [7,8], and product pricing [9]. Meanwhile, the restriction of quasi-strong monotonicity in minimization problems such as one-point convexity [10] is also used in analyzing neural networks.
> > >
> > > In Zhou et al. (2017) [5], the strongly coherent optimization problems have been studied, which are special cases of the quasi-strongly monotone VI. Several examples of problems satisfying this assumption appeared in [5]. As we mentioned in our original response, the assumption of quasi-strong monotonicity is equivalent to the concept of strong stability described in [32] (see [32] for more examples of these problems).
> > >
> > > As a more concrete min-max application, we refer the reviewer to [11] where two-agent zero-sum reinforcement learning problems are studied. There, the authors consider a special case of the general two agent zero-sum RL problem, called von Neumann’s ratio game. Following the discussion related to this appeared in [2] one can potentially construct examples of the von Neumann ratio game where the quasi-strong monotonicity is satisfied.
> > >
> > > Having mentioned all of the above papers (with examples and applications) that use similar conditions to our setting, we highlight again that our work is primarily theoretical. We believe that our results will help in closing the gap between the stochastic optimization algorithms and methods for solving stochastic games (stochastic VI problems) and can open up many avenues for further development and research in both areas.
> > >
> > > **References mentioned above related to applications:**
> > >
> > > [1] Daskalakis, C., Skoulakis, S., and Zampetakis, M. (2021). The complexity of constrained min-max optimization. In Proc. STOC’21.
> > >
> > > [2] Diakonikolas, J., Daskalakis C, and Jordan M. "Efficient methods for structured nonconvex-nonconcave min-max optimization." In AISTATS 2021.
> > >
> > > [3] Song, C., Zhou, Y., Zhou, Z., Jiang, Y., and Ma, Y. (2020). Optimistic dual extrapolation for coherent non-monotone variational inequalities. In Proc. NeurIPS’20.
> > >
> > > [4] Kannan A. and Shanbhag U.V. (2019). Optimal stochastic extragradient schemes for pseudomonotone stochastic variational inequality problems and their variants. Computational Optimization and Applications, 74(3):779–820, 2019.
> > >
> > > [5] Zhou, Z., Mertikopoulos, P., Bambos, N., Boyd, S. and Glynn, P.W., (2017). Stochastic mirror descent in variationally coherent optimization problems. In Proc. NeurIPS’17.
> > >
> > > [6] Luigi Brighi and Reinhard John. Characterizations of pseudomonotone maps and economic equilibrium. Journal of Statistics and Management Systems, 5(1-3):253–273, 2002.
> > >
> > > [7] Alexandre Mikhajlovich Elizarov and AN Kalimullina. Maximization of the lift/drag ratio of airfoils with a turbulent boundary layer: Sharp estimates, approximation, and numerical solutions. Computational Mathematics and Mathematical Physics, 49(3):559–572, 2009
> > >
> > > [8] Aymeric Rousseau, Phil Sharer, Sylvain Pagerit, and Sujit Das. Trade-off between fuel economy and cost for advanced vehicle configurations. In 20th International Electric Vehicle Symposium (EVS20), Monaco, 2005
> > >
> > > [9] S Chan Choi, Wayne S DeSarbo, and Patrick T Harker. Product positioning under price competition. Management Science, 36(2):175–199, 1990
> > >
> > > [10] Yuanzhi Li and Yang Yuan. Convergence analysis of two-layer neural networks with relu activation. In Proc. NeurIPS’17.
> > >
> > > [11] Daskalakis, C., Foster, D. J., and Golowich, N. (2020). Independent policy gradient methods for competitive reinforcement learning. In Proc. NeurIPS’20.

---

> ### Author Response · Authors · 2021-08-10
> **All issues handled**
>
> We would like to thank the reviewer for their time.
>
> The main concerns of Rev.kKUx is:
> * (i) on the setting under study and
> * (ii) on the importance/novelty of our result.
>
> We address all issues raised below.
>
> In our opinion all issues raised can easily be handled in the camera-ready version by providing more details on the setting under study (see response below) and clarifying more the statements that the reviewer characterized as overclaims. Thus, we do not believe a score of 4 is appropriate.
>
> We believe that our work provides strong theoretical results that can open up many avenues for further development and research.
>
> If you agree that we managed to address all issues, please consider raising your mark. If you believe this is not the case, please let us know so that we have a chance to respond.

---

### Official Review · Reviewer_PL4z · 2021-07-19

**Rating:** 4
**Confidence:** 4

**Summary:**

This paper proposes convergence analysis on the optimization algorithms of smooth games, where the problem is formulated as a stochastic variational inequality, under the novel assumptions called expected co-coercivity and quasi-strong monotonicity. These assumptions serve the role of smoothness and strong convexity in the classical analyses of SGD for convex minimization. The authors show the last-iterate convergence rates for stochastic gradient descent-ascent (SGDA) and stochastic consensus optimization (SCO). The authors also relax the assumptions on the coordinate sampling schemes and on the noise.

**Main Review:**


[Strengths]
This work introduces a new assumption and provides novel analyses for SGDA and consensus optimization.
- Overall, the paper is well-written and states its contribution clearly.

[Weaknesses]
- While the results of this work are new, the proof techniques very closely follow the prior work of Gower et al (2019), which provided the convergence analysis on stochastic GD under 'expected smoothness' assumption. Expected co-coercivity for operator setting is analogous to the expected smoothness for nonconvex optimization setting, as these two assumptions only differ in the expression on the upper bound of the squared gradient norm. The proof structures of the work are even quite similar; both works provide the lemma of the similar form which plays essential role in the proof (Lemma 2.4 in Gower et al (2018) & Lemma 3.4 in this paper), prove the main claim in essentially the same manner, using the similar choice of step sizes.
- I am not sure that I agree with the authors that the introduction of expected cocoercivity is a significant contribution. For that to be the case, I feel like the authors should not only show how the assumption can be used to prove things (which the authors do) but should also demonstrate why the assumption is realistic or interesting (which I feel the authors do not adequately do).
- Despite the authors' counter example of A.6, I am not convinced that the quasi-strong monotonicity is an assumption significantly or meaningfully weaker than the usual strong monotonicity (strong convex-concavity).
 - It is unclear to me how the "VI" perspective meaningfully interacts with the proofs or the theory.



Overall, while the results presented in this paper are useful to have, I feel that the analysis is too straightforward of an extension of the Gower et al. 2019 prior work. I was not able to find any novel proof techniques. For these reasons, I do not think this paper meets the bar of a NeurIPS publication.


[Minor Issues]
- p.4, line 124 (typo) 'uniforms boundedness' -> 'uniform boundedness'
- EC(l_\xi) is also dependent on the distribution \mathcal{D}. Including \mathcal{D} in the notation of EC may clarify the definition.


**Time Spent Reviewing:**

4

---

> ### Author Response · Authors · 2021-08-10
> **Addressing the comment on “VI” perspective in proofs**
>
> **The comment of the reviewer:**
>
> “It is unclear to me how the "VI" perspective meaningfully interacts with the proofs or the theory.”
>
> **Our response:**
>
> The vector field in VI problem is not a gradient. Thus, there is no notion of function value and thus no descent lemma. As a result one is not able to use expected smoothness (ES) or similar conditions as assumptions to provide any convergence guarantees (the upper bound requires the function suboptimality). The proposal and use of expected co-coercivity (EC) is one of the main points of how the VI perspective meaningfully interacts with the proofs and the proposed theory.
>
> At this point, let us mention again that the upper bound of ES in Gower et al. 2019 requires function suboptimality (a concept that cannot be useful in VI problems), while EC is a strict generalization that applies to both optimization and VI problems.
>
> As we argue in the paper, expected co-coercivity is a natural notion for an efficient analysis of stochastic algorithms for solving VI problems. Out of all known sets of assumptions, EC yields meaningful and realistic convergence guarantees without requiring extra strong assumptions like bounded noise or bounded variance.

---

> ### Author Response · Authors · 2021-08-10
> **On quasi-strong monotonicity**
>
> Quasi-strong monotonicity as an assumption is weaker than strong monotonicity as it includes some non-monotone games as special cases. In optimization literature the class of  quasi-strongly convex functions is a rich class of problems and several papers proposed convergence guarantees in this setting. Quasi-strong monotonicity is the generalization of this class of problems in VI setting.
> Note that quasi-strong monotonicity has already been used in other works and is related to the strong stability condition (see [32], P. Mertikopoulos and Z. Zhou. "Learning in games with continuous action sets and unknown payoff functions". Mathematical Programming. 173: 465–507. 2019.)
>
> Similar structured non-monotone classes of VI problems  (which include strongly monotone problems as a special case) have been considered in other recent works. For example,
> *  an “error bound condition” is used in the analysis of stochastic extragradient in
>
> [21] Y.-G. Hsieh, F. Iutzeler, J. Malick, and P. Mertikopoulos. Explore aggressively, update conservatively: Stochastic extragradient methods with variable stepsize scaling. In NeurIPS, 2020.
>
> *  and a “Two-sided Polyak- Lojasiewicz inequality” is assumed in:
>
> [50] J. Yang, N. Kiyavash, and N. He. Global convergence and variance-reduced optimization for a class of nonconvex-nonconcave minimax problems. NeurIPS, 2020.
>
> **Remark:** Let us also highlight that even if we focus on strongly monotone operators (special case of quasi-strong monotonicity), our results would still be novel and theoretically interesting. Even for this classical setting, our work is the first that provides last-iterate convergence guarantees of SGDA and SCO without requiring the extra bounded noise or bounded variance assumptions (for the SCO there were no known convergence guarantees before our work). Note that in Proposition A.7 of Appendix A.5 we prove that assuming strong monotonicity and bounded noise leads to a contradiction (empty set of operators).\
>  In other words, the goal of our work is to provide tight convergence guarantees for two of the most prominent algorithms for solving VI problems (SGDA and SCO) in a setting where this is possible. We believe that our work will work as a first step towards a better understanding of the behaviour of these methods in general non-monotone problems (which is the ultimate goal).

---

> ### Author Response · Authors · 2021-08-10
> **Significance of expected co-coercivity**
>
> As the reviewer mentioned for the EC to be a significant contribution we should have shown that EC:
> * (i) “can be used to prove things (which the authors do)“ and
> * (ii)  “demonstrate why the assumption is realistic or interesting (which I feel the authors do not adequately do).”
>
> Regarding point (ii), in Section 3.1 of our paper we provide Proposition 3.5 and 3.6 that precisely show that EC is not an assumption but a condition satisfied in several practical scenarios. These propositions show exactly that EC is a realistic and interesting condition. For example, when $\xi_i$ are co-coercive around $x^*$ (Proposition 3.5) or when they are Lipschitz continuous and $\xi$ be quasi strongly monotone (Proposition 3.6) the EC is always satisfied (is not an assumption but a property of the problem under study). This allows us to use EC in our analysis without any extra assumption like bounded noise or bounded variance that was previously used in the analyses of stochastic algorithms for solving stochastic VI problems.
>
> We should also highlight that EC is a condition that combines both the properties of the user-defined distribution $\cal D$ and the properties of operator $\xi$. This in combination with the arbitrary sampling regime (see Section 2) allows us to capture and analyze all possible samplings for the two algorithms under study (SGDA and SCO). As we mentioned in our work, an analysis under the arbitrary sampling paradigm was not proposed before for algorithms solving VI problems.

---

> ### Author Response · Authors · 2021-08-10
> **On main contributions and proof techniques**
>
>  ## **Expected co-coercivity Vs Expected smoothness:**
>
> Expected co-coercivity for operator setting is indeed analogous to the expected smoothness for nonconvex smooth optimization setting. In particular, Expected co-coercivity (EC)  for operator setting is a strict generalization of the expected smoothness (ES) previously used in the optimization literature (Gower et al. 2019). This is clearly a benefit of our work. The reviewer mentions that “the two assumptions only differ in the expression on the upper bound of the squared gradient norm”.  To this end, we note that the upper bound of ES in Gower et al. 2019 requires function suboptimality (a concept that cannot be useful in VI problems), while EC is a strict generalization that applies to both optimization and VI problems. Thus, the **difference of the two notions does not solely lie on the norm bound, but begins at the deeper, conceptual level.**
>
> ## **On the convergence proofs for SGDA:**
>
> We agree with Rev. PL4z that the proof techniques for our analysis of SGDA  follow the work of Gower et al. 2019 (however under different and more general assumptions) since in our opinion this is the more natural way of presenting the convergence guarantees. We could have easily presented our contributions using a different approach and not highlight the connection to Expected smoothness in such an obvious way. We did not do that on purpose.  We want the paper to be as clear as possible and make the connections to the optimization literature easy to follow. Table 1 serves exactly this purpose. We also highlight that the work of Gower et al. 2019, focuses only on standard optimization problems and has no results on games or variational inequalities (which is a more challenging task with several open questions).
>
> For the analysis of SGDA, having novel proof techniques is not the main selling point of our work. As we clearly highlight in the paper, using recently developed optimization tools we were able to come up with a reasonable condition for the VI problems (EC assumption) and provide the **first last-iterate convergence guarantees of SGDA for the setting under study** (quasi-strongly monotone operators), without requiring strong assumptions like bounded noise and bounded variance that previous works assumed. We were also able to provide the **first analysis of SGDA under the arbitrary sampling paradigm** that also includes the deterministic gradient descent ascent as a special case. To the best of our knowledge the **step-sizes proposed in our theorems for the convergence of SGDA are also novel** and have not been considered before.
>
> ## **On the convergence proofs for SCO:**
>
> We find it unfair that the convergence guarantees of SCO, which is one of the most important contributions of our work, is not mentioned at all by Rev. PL4z. Note that SCO is a lightweight second order method, previously used for training GANs and which up to this point has no convergence guarantees . In our work, using **novel proof techniques** and assumptions,  we prove the **first last-iterate convergence for SCO** (there were no known convergence guarantees of SCO before our work), for both constant and decreasing step-sizes. As a corollary of our results, we obtain an **improved convergence analysis for the deterministic CO**. Furthermore, we explain how the update rule of the stochastic Hamiltonian gradient descent (Loizou et al., 2020) is a special case of the SCO and shows that in this scenario, **our analysis matches the theoretical guarantees presented in Loizou et al. (2020)**. We show that for the convergence of SCO the quasi-strong monotonicity is not even necessary and we provide convergence guarantees under the variational stability condition.
>
> -------------------
> **[On final comment on novelty of proof techniques]**
>
> Reviewer PL4z, mentions the following at the end of the review: *“Overall, while the results presented in this paper are useful to have, I feel that the analysis is too straightforward of an extension of the Gower et al. 2019. I was not able to find any novel proof techniques. For these reasons, I do not think this paper meets the bar of a NeurIPS publication.”*
>
> We respectfully stand by our claim of novelty (please check our full response) and we politely disagree with the reasoning of the above statement. In our opinion, the fact that part of the proof techniques is an extension of previous works is no reason for suggesting rejection of a paper.

---

> ### Author Response · Authors · 2021-08-10
> **All issues handled**
>
> Thank you for investing time in the review process and for the detailed review.
>
> We are glad that you find our paper well-written and that it states its contribution clearly. Thank you for acknowledging that the theoretical results are new and that the analyses of SGDA and SCO are novel.
>
> In our response we are able to handle all issues raised related to:
> * Main contributions and proof techniques
> * Significance of Expected co-coercivity
> * Quasi-strong monotonicity
> * “VI” perspective in proofs and theory
>
> As we explain in more detail below, with our approach we answer several open questions in the area and provide important last-iterate convergence guarantees for two of the most prominent algorithms for solving unconstrained smooth games (unconstrained  VI problems). We feel that our work opens up many avenues for further development and research and makes clear connections between different areas of research (stochastic optimization algorithms and methods for solving stochastic VI problems) that could be very helpful for the community.
>
> If you agree that we managed to address all issues, please consider raising your mark. If you believe this is not the case, please let us know so that we have a chance to respond.

---

### Author Response · Authors · 2021-08-10
**Thanks to all reviewers**

Thanks to all reviewers for examining our manuscript and providing valuable suggestions.

We address all raised issues below.

We refer all reviewers to the comments of Reviewer LrZq, as they greatly describe the significance of our work and our main contributions.

Here we highlight again that with our work:
 * We propose the **expected co-coercivity (EC)** condition,  explain its benefits and show that it is strictly weaker than the bounded variance assumption and “growth conditions” previously used for the analysis of stochastic algorithms for solving VI problems.

* Using the EC, we prove the **first last-iterate convergence guarantees of SGDA** in the setting under study (quasi-strongly monotone games) without any unrealistic noise assumption.

 * Using the EC, we provide the **first convergence analysis of a stochastic variant (SCO) of the consensus optimization**. In particular, we prove last-iterate convergence for SCO, for both constant and decreasing step-sizes. As a corollary of our results, we obtain an improved convergence analysis for the deterministic CO. Furthermore, we explain how the update rule of the stochastic Hamiltonian gradient descent (Loizou et al., 2020) is a special case of the SCO and shows that in this scenario, our analysis matches the theoretical guarantees presented in Loizou et al. (2020).

 * We give the **first stochastic reformulation of the variational inequality problem** which enables us to provide convergence guarantees of SGDA and SCO under the **arbitrary sampling paradigm**. This allows us to give insights into the complexity of minibatching. In other words, our convergence guarantees of SGDA and SCO hold for all choices of proper sampling as we explain in Appendix A.3 (this is the first time that such analysis takes place for stochastic methods solving VI problems).

**We hope that you will engage with us in a back-and-forth discussion and we will be most happy to answer any remaining questions.**

---

### Decision · Program_Chairs · 2021-09-27

**Decision:**

Accept (Poster)

**Comment:**

This paper introduces a new expected co-coercivity condition, explain its benefits, and provide the first last-iterate convergence guarantees of SGDA and SCO under this condition for solving a class of stochastic variational inequality problems that are potentially non-monotone. Reviewers find the new conditions very interesting, and very mild in many settings. Therefore the new convergence results under these mild settings are rather strong. The techniques developed here are also of broad interests.